# Evaluation of a Resilience-Driven Operational Concept to Manage Drone Intrusions in Airports

**Domenico Pascarella** [1],*, **Gabriella Gigante** [1], **Angela Vozella** [1], **Maurizio Sodano** [2], **Marco Ippolito** [2], **Pierre Bieber** [3], **Thomas Dubot** [3] and **Edgar Martinavarro** [4]

1. Safety and Security Department, CIRA—Italian Aerospace Research Centre, Via Maiorise, 81043 Capua, Italy; g.gigante@cira.it (G.G.); a.vozella@cira.it (A.V.)
2. Soul Software—Italian Software Factory, Viale S. Modestino 33, 83013 Mercogliano, Italy; maurizio.sodano@soulsoftware.eu (M.S.); marco.ippolito@soulsoftware.eu (M.I.)
3. ONERA/DTIS, Université de Toulouse, BP 74025, 31055 Toulouse Cedex, France; pierre.bieber@onera.fr (P.B.); thomas.dubot@onera.fr (T.D.)
4. Unmanned Aerial Platforms Department, INTA—National Institute of Aerospace Technology, 28850 Torrejón de Ardoz, Spain; martinavarro@inta.es
* Correspondence: d.pascarella@cira.it

**Abstract:** The drone market's growth poses a serious threat to the negligent, illicit, or non-cooperative use of drones, especially in airports and their surroundings. Effective protection of an airport against drone intrusions should guarantee mandatory safety levels but should also rely on a resilience-driven operational concept aimed at managing the intrusions without necessarily implying the closure of the airport. The concept faces both safety-related and security-related threats and is based on the definitions of: (i) new roles and responsibilities; (ii) a set of operational phases, accomplished by means of specific technological building blocks; (iii) a new operational procedure blending smoothly with existing aerodrome procedures in place. The paper investigates the evaluation of such a resilience-driven operational concept tailored to drone-intrusion features, airport features, and current operations. The proposed concept was evaluated by applying it to a concrete case study related to Milan Malpensa Airport. The evaluation was carried out by real-time simulations and event tree analysis, exploiting the implementation of specific simulation tools and the assessment of resilience-oriented metrics. The achieved results show the effectiveness of the proposed operational concept and elicit further requirements for future counter-drone systems in airports.

**Keywords:** drone intrusion; airport operations; resilience assessment; real-time simulation; event tree analysis; counter-drone systems





## 1. Introduction

Airports have become critical transportation infrastructures, representing the key nodes of the aviation system network with strict interdependencies between them, vehicles, and people. Operational issues at a single airport may impact the entire National Airspace System (NAS) and may trigger cross-border or worldwide ripple effects. The closedown of a large or medium-sized airport reverberates across the NAS, resulting in: (i) the immediate rerouting of hundreds of arrivals; (ii) the delay or the cancellation of thousands of domestic and international flights; (iii) cascading traffic disruptions, possibly affecting not only the specific day but also the course of the following days.

Tackling these airport issues requires overcoming traditional performance-driven approaches for the design of new solutions within air traffic management (ATM). Indeed, such approaches always apply the safety-as-a-priority paradigm by mainly focusing on system failures causing catastrophic or major accidents, which are events with large safety impacts. More generally, beyond these events, other ATM adverse events may push the dynamics of ATM far away from its point of operation (even without safety impacts)

and dramatically affect the performance and economic aspects of the system [1]. The solutions to these adverse events have to consider the system as a whole, involving all of its dimensions, i.e., hardware, software, human, and environment. Accordingly, the systematic implementation of these solutions is being addressed by introducing the resilience concept into the ATM domain.

EUROCONTROL defines resilience as "the intrinsic ability of a system to adjust its functioning prior to, during, or following changes and disturbances, so that it can sustain required operations under both expected and unexpected conditions" [2]. This definition is mainly based on the work of Hollnagel, who has also generally addressed resilience as "the ability of a system or an organization to react to and recover from disturbances at an early stage, with minimal effect on the dynamic stability" [3]. Unlike robustness, a resilient ATM system responds to a reference disturbance within a limited time horizon by transient perturbation, i.e., by modifying its functioning [4].

Recent episodes highlight the increasing demand for resilient airports to ensure that the NAS is no longer vulnerable to disruptive events that occur at a single airport [5]. As an example, one may mention the failure of the electrical system at the Hartsfield-Jackson Atlanta International Airport in 2017, which shut down the airport for several hours and caused a loss of upwards of USD 50 million in revenue for the tenant airlines [5]. The resilience of an airport to an adverse event should exploit three phases [5]: (i) the "before" phase, for "resisting" through prior prevention and mitigation planning; (ii) the "during" phase, for "absorbing" through prior response planning; (iii) the "after" phase, for adapting through prior recovery planning. Clearly, proper implementation of these phases involves different dimensions of the planning process, i.e., technological, organizational, social, and economic.

However, in the last five years, the explosion of the drone market has brought out another serious threat against the resilience of airports related to the negligent, illicit, or non-cooperative use of drones. One of the well-known examples is represented by the drone intrusion at London Gatwick Airport in December 2018, during which an unknown number of overflying drones caused a 33 h paralysis of airport operations [6]. For this Gatwick episode, the European Union Aviation Safety Agency (EASA) has estimated a cost of EUR 64 million [7].

Moreover, EASA has published a counter-drone action plan [8] and some guidelines [7,9,10] in which counter-drone systems are recommended to mitigate drone intrusion impact on the airport ecosystem. These systems are also named counter-UAS or C-UAS systems, where the acronym UAS stands for unmanned aerial system. Thus, several C-UAS systems are being developed to satisfy the growing need to defend against intruding drones, especially at low altitudes and in the operational envelope of drones. Nevertheless, in the case of airport operations, these systems usually exploit a policy based on reactive countermeasures, consisting of the detection of the intrusion and of the subsequent closure of the overall airport until the threat has reduced. On the one hand, such a policy guarantees the safety levels of the airports in case of drone intrusions and is in line with the safety-as-a-priority paradigm; on the other hand, it significantly penalizes the other performance levels of the airports since it implies relevant disruptions for the overall traffic network. In other words, airport drone intrusions are a glaring example of how the decoupled usage of safety aspects and of the other performance aspects may lead to an ultra-safe ATM, whose safety would conflict with the other performance dimensions (e.g., capacity, economy, time performance, human performance, etc.).

The only way to protect the resilience of the airports against drone intrusions is to build an effective drone intrusion management system (DIMS), which should deploy the three phases (before, during, and after) to allow the airport to resist, absorb, and adapt against the intrusion. For this purpose, in addition to reactive countermeasures, the DIMS should exploit both preventive countermeasures and proactive countermeasures. For example, among the preventive countermeasures, there are the deterrence and the denial to enter protected areas. Instead, proactive countermeasures are mainly based on: (i) creating a

complete and timely situational awareness about drone intrusions in order to support the dynamic assessment of their risks; (ii) designing procedures and protocols to manage the intrusions in order to mitigate their impact as much as possible.

The ASPRID (Airport System PRotection from Intruding Drones) project [11,12] is an exploratory research project, funded by the Single European Sky ATM Research (SESAR) program, to develop an operational concept for the protection of airports against drone intrusions in a resilient way. The operational concept is based on proactive countermeasures and aims at increasing the resilience of the airport (that is, minimizing the performance degradation) against drone intrusions by avoiding the closure of the overall airport if not strictly necessary. This results in: (i) limiting the interruption to those operations that are strictly affected by the intruder; (ii) enabling "chirurgical" actions for mitigation and response, considering the closure of the airport as the last option. According to the phases and the dimensions for airport resilience in [5], the operational concept was designed with reference to:

- The "before" and "during" phases for "resisting" and "absorbing" the intrusion, in order to increase the resilience of the airport by means of mitigation and response planning;
- Technological and organizational dimensions in order to address a process model and an operational procedure for a proactive DIMS, employing different building blocks, from the detection up to the mitigation systems.

Our previous work in [13] puts forward the basis for a systematic process of risk management within the DIMS of an airport, describing a quantitative assessment of the historical features of drone intrusions in airports. Our other work in [14] describes a unified methodological framework for the risk assessment of airport drone intrusions, combining model-based and data-driven specifications. Leveraging the risk assessment results in [13, 14], this paper investigates the evaluation of the proposed resilience-driven operational concept for the DIMS, tailored on drone-intrusion features, airport features, and current operations. The concept was evaluated by applying it to a concrete case study related to Milan Malpensa Airport, in line with the case study described in [14]. The evaluation was carried out by real-time simulations and event tree analysis (ETA), exploiting the implementation of a specific simulation platform and the assessment of resilience-oriented metrics. The achieved results show the effectiveness of the proposed operational concept and elicit further requirements for future counter-drone systems in airports.

The article is organized as follows. Section 1 presents a brief overview of the issues for airport resilience related to drone intrusions, highlighting DIMSs as a proper tool for resilient protection. Section 2 analyzes the related work for both airport resilience and counter-drone systems in airports. Section 3 describes the proposed operational concept. Section 4 reports the methodology for the resilient-driven evaluation of the operational concept, including the description of the case study. Section 5 illustrates the evaluation results. Section 6 provides a discussion of the proposed operational concept, based on the achieved results. Section 7 provides the conclusions. Appendix A reports the detailed results of the simulation activities.

## 2. Related Work

This section describes the related work about airport resilience and counter-drone systems for airports.

### 2.1. Airport Resilience

In transportation networks, research on resilience mainly faces natural hazards or extreme weather events [15]. For the specific case of the air transport system, research on resilience mostly focuses on analyzing the complex systems within ATM and air traffic networks by means of complexity science in order to model the resilience of the system or network and to identify the most critical elements [16].

Reference [15] lists some European projects that have provided systematic approaches for the quantitative assessment of resilience and for a resilience-driven development of

ATM solutions. For example, one mentioned project is Resilience2050 [17], which has developed models to support mathematical analyses of resilience in ATM scenarios, both in normal operations as well as disturbances [18]. The project considered disturbances related to weather conditions, bad visibility issues, runway configuration changes, and staffing problems [19].

Another mentioned project is the SAFECORAM (Sharing of Authority in Failure/ Emergency COndition for Resilience of Air traffic Management) project, which identified a model to evaluate resilience in terms of performance degradation in failure/emergency conditions and developed a concept to optimize the resilience index of an ATM system by means of task allocation and authority sharing between humans and automated components [20]. The concept was tested in some emergency scenarios, such as the aircraft's uplink loss during the en-route phase [21].

Thus, our proposed work represents the first attempt at explicitly evaluating airport resilience for threat scenarios related to drone intrusions.

### 2.2. Counter-Drone Systems for Airports

This subsection discusses the counter-drone systems for airports by describing both the current regulations and the research works.

The regulation of drone intrusions in airport environments has become a pressing issue in recent years, as the number of drones in operation continues to grow and the potential risks they pose to aviation safety become increasingly clear. In order to address this issue, several initiatives have been undertaken by key regulatory and standardization bodies, such as EASA and the European Organisation for Civil Aviation Equipment (EUROCAE), and industry associations, such as Airports Council International (ACI).

EASA formed the counter-UAS task force in November 2019 to work on objective 2 of the EASA C-UAS action plan [8], which is preparing aerodromes to mitigate risks from unauthorized drone use. The task force delivered a manual, named Drone Incident Management Manual for Aerodromes, which provides guidance on the management of drone incidents in and around airport environments. This manual is divided into three parts: Part 1 [7], which is publicly accessible, provides an overview of the challenges posed by unauthorized drones in airport environments; Part 2 [9] and Part 3 [10], which are more restricted in distribution due to their sensitive nature, provide guidance and recommendations for incident management and practical tools for implementation.

Within the same timeframe, EUROCAE and the Radio Technical Commission for Aeronautics (RTCA) jointly developed an Operational Services and Environment Definition (OSED), named ED-286 [22]. This document provides a detailed description of the operational services of a C-UAS system and the environment in which such a system will operate. It proposes operational requirements and associated assumptions and defines the following capabilities: global detection, decision support, neutralization, and data collection.

Finally, ACI formed a drone task force in February 2019 with the goal of discussing drone operations at airports with all relevant stakeholders, not only to exchange information but also to provide guidance on how drone operations could be facilitated whilst ensuring the necessary safety and security levels at airports. The task force created the Concept of Operations (ConOps) [23], which specifies a set of recommendations for the airport community (airport operators, air control, surveillance units, drone operators, etc.).

All these initiatives were created to provide a comprehensive framework for the management of drone incidents in airport environments. Together they represent a significant step forward in the regulation of drone intrusions in airport environments and helped to ensure the safety and security of passengers and personnel. Nevertheless, they require local implementations at the level of member states, depending on national regulatory specificities.

Concerning research works, reference [24] studied drone incidents in the vicinity of worldwide airports to deliver quantitative and qualitative analyses, proposing possible mitigation measures. Reference [25] describes an action plan for airport stakeholders to

defend an airport against misused drones. Lastly, several works provide a detailed analysis of the technological options for counter-drone systems, e.g., [25–27].

Leveraging the available regulations and C-UAS technological options, our proposed work also represents the first attempt at defining an operational concept for airport protection against drone intrusion, specifically designed to manage the resilience of airport operations.

### 2.3. Research Gaps and Proposed Innovation

This subsection provides a summary of the related work by pointing out the research gaps. In addition, it highlights the proposed innovation of this work.

In regard to airport resilience (Section 2.1), previous research was mainly related to the quantitative assessment of resilience against specific disturbances and off-nominal ATM scenarios, such as those related to weather conditions, bad visibility issues, runway configuration changes, staffing problems, and failures of the overall transportation system. Several works have developed resilience metrics and models by applying complexity science. To the best of our knowledge, no prior work has defined and assessed airport resilience for threat scenarios related to drone intrusions.

In regard to counter-drone systems for airports (Section 2.2), both regulations and research works have to be considered. The former provides a comprehensive framework for the management of drone incidents in airport environments. The latter mainly addresses safety analyses and deals with detailed technological options for "isolated" capabilities of counter-drone systems, e.g., in terms of detection and neutralization. To the best of our knowledge, no prior work has: (i) developed an "integrated" concept for airport protection against drone intrusions; (ii) considered resilience features for the design of counter-drone operations.

Based on the aforementioned research gaps, this work proposes a twofold innovation. On the one hand, it assesses the airport's resilience for threat scenarios related to drone intrusions by introducing explicit resilience metrics and by defining a resilience-driven evaluation methodology of counter-drone solutions. On the other hand, it designs an integrated and resilience-driven operational concept for drone intrusion management in airports by: (i) leveraging the available regulations and C-UAS technological options; (ii) managing both accidental and malicious intrusions; (iii) minimizing the airport's performance degradation.

## 3. Operational Concept for Drone Intrusion Management

This section describes ASPRID's operational concept, i.e., the proposed operational concept for the DIMS. Note that the operational concept "is designed to give an overall picture of the operations using one or more specific systems, or a set of related systems, in the organization's operational environment from the users' and operators' perspective" [28]. Thus, in our case, the ASPRID operational concept is a user-oriented and operator-oriented description of the operations and the building blocks that implement the DIMS. Such a description includes:

- ASPRID's process model, which specifies the actors and the workflow of the operations performed by the actors to accomplish the management of drone intrusions;
- ASPRID's operational procedure, which specifies the step-by-step instructions for the operators to accomplish the management of drone intrusions.

The operational concept considers both safety-related (i.e., accidental) and security-related (i.e., malicious) intrusions. In the case of security-related intrusions, these represent deliberate attacks, which may include both drone physical intrusions and drone cyber intrusions (i.e., cyberattacks using drones as cyber weapons) [14]. However, for the purposes of the ASPRID operational concept, only physical intrusions by single drones (i.e., not in teams or swarms) are considered as a reference threat in the stated problem.

The following subsections report the ASPRID process model and operational procedure.

*3.1. Process Model*

The ASPRID process model relies on the implementation of the roles and responsibilities of the actors in the Drone Incident Management Cell (DIMC), which refers to a group that responds to unauthorized drone activity in the airport environment, as recommended by EASA guidelines [7]. Main roles, such as air traffic controllers (ATCOs), including both tower controllers and ground controllers, or law enforcement authorities (LEAs), are refined to consider the management of drone intrusions according to: (i) regulation principles; (ii) the inclusion of the new actor for ASPRID's operational concept, i.e., the ASPRID system.

In detail, the ASPRID system is a technological actor performing the following functions:

1.　The detection and tracking of unauthorized drones;
2.　Alerting to drone intrusions;
3.　The classification and identification of unauthorized drones, i.e., reconnaissance of drone features (type, speed, etc.);
4.　Providing support to threat assessments, i.e., the estimation of the level of risk related to drone intrusion (e.g., as a function of the type and the speed of the drone, of the current ongoing operations, etc.);
5.　Providing support to C-UAS actions, i.e., the neutralization of unauthorized drones.

Thus, the ASPRID system provides the human actors with full situational awareness of the intrusion. Note that the first three functions are autonomous, whereas the last two are intended as support for the decisions of a human actor. Clearly, the required equipment for these functions is a design choice depending on the specific airport (some preliminary information is available for the simulation of the case study in Section 4.2.3). Moreover, the ASPRID system introduces a new human actor, named the ASPRID operator, who is responsible for supervising and controlling the ASPRID system.

The process model clarifies how the ASPRID system could support each DIMC subprocess and the associated communication exchanges. Three different phases may be distinguished:

- The strategic phase, during which roles and responsibilities are defined, tools are trained and configured, and main scenarios are simulated;
- The tactical phase, during which all actors manage the threat according to procedures defined in the strategic phase, the latest information data, and the ASPRID analysis;
- The post-operations (post-ops) phase, which is used to investigate past incidents and how processes may be improved.

Within the context of major airports, the ASPRID system should provide 24/7 situational awareness to all users by uninterrupted monitoring of the airport environment, making the tactical phase a rolling process that is always active. Figure 1 shows the flowchart for the main subprocesses of the tactical phase, using the Business Process Model and Notation (BPMN) [29]. It is divided into four swimlanes, representing the following actors or responsibilities: the DIMC, the ASPRID system (including ASPRID human roles that are detailed in Section 3.2), aircraft operators in the jurisdiction of the airport or air traffic control (ATC), and other external actors such as U-space. The DIMC lane itself is subdivided into three sub-lanes to distinguish actions realized within the DIMC by the aerodrome's safety and security operators, ATCOs and LEA. Boxes inside lanes represent activities under the responsibility of each actor, and plain arrows indicate the sequence of actions. Finally, dotted lines represent information exchanges between actors.

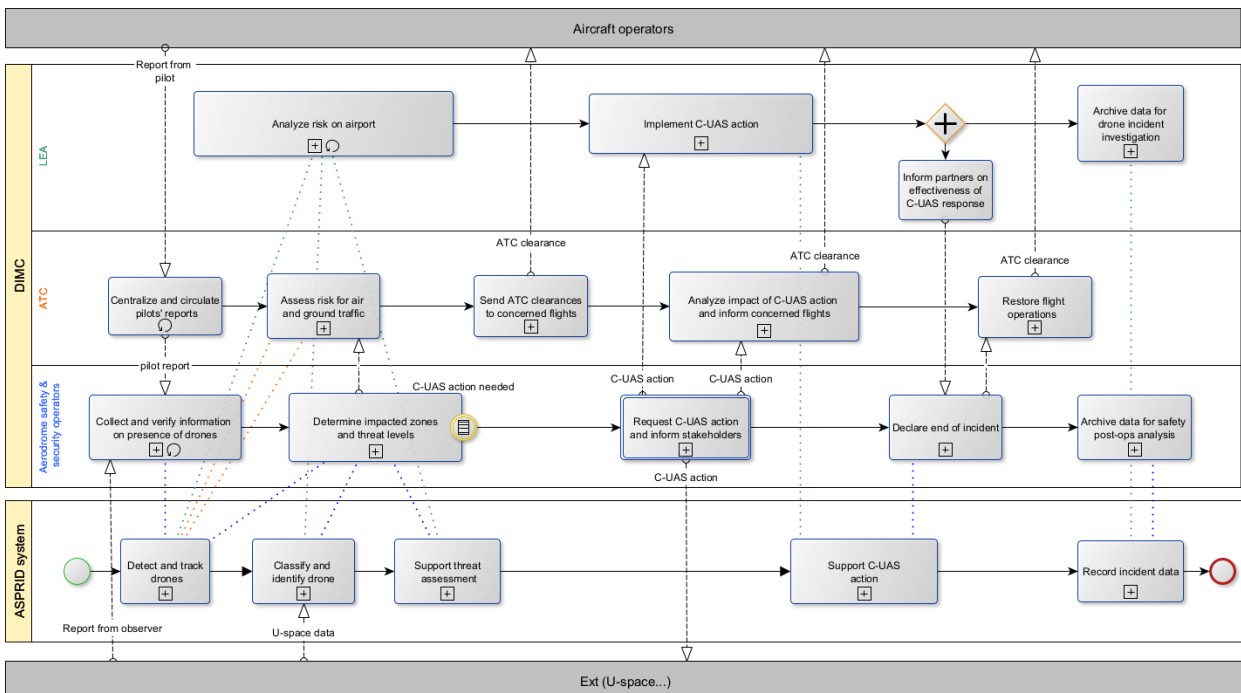

**Figure 1.** BPMN process diagram of the tactical phase of ASPRID process model.

*3.2. Operational Procedure*

ASPRID's operational procedure relies on the definition of critical assets of the airports. In detail, a critical asset (or critical element or, simply, asset) is any airport asset that exhibits a weakness against drone intrusions, i.e., the intrusion may result in a temporary or permanent interruption of the airport's operations involving the asset [14]. Thus, critical assets are those that are susceptible to mishap risks related to drone intrusions and are identified by means of a threat analysis within the risk assessment [14], which is part of the strategic phase of the ASPRID process model, as addressed in Section 3.1. Note that the assets may be both stationary and mobile (e.g., an aircraft).

For each asset, the threat analysis also defines a safety radius, which represents the minimum physical distance between the asset itself and any intruding drone in order for the asset to continue to operate under nominal conditions without interruption. Thus, the safety radius is intended as a safety bubble around the asset with different geometrical shapes, not necessarily circular. For example, runways are better associated with polygonal safety bubbles.

Figure 2 shows the principles for the operational management of a generic critical asset. Each asset is associated with the following alert levels in the case of drone intrusion:

- White level—The drone is outside of the safety radius. The asset operates in nominal mode.
- Yellow level—The drone is within the safety radius but does not physically interfere with the asset's operations. The asset operates in nominal mode but requires a high monitoring level.
- Orange level—The drone is within the safety radius and physically interferes with the asset's operations, but the collision between the drone and the asset is not imminent. The asset operates in degraded mode. Interdependent assets may also operate in degraded mode.
- Red level—The drone is within the safety radius, it physically interferes with the asset's operations, and the collision between the drone and the asset is imminent. The asset operates in suspended mode, i.e., it requires a shutdown for its operations. Interdependent assets or the full airport may also require a shutdown.

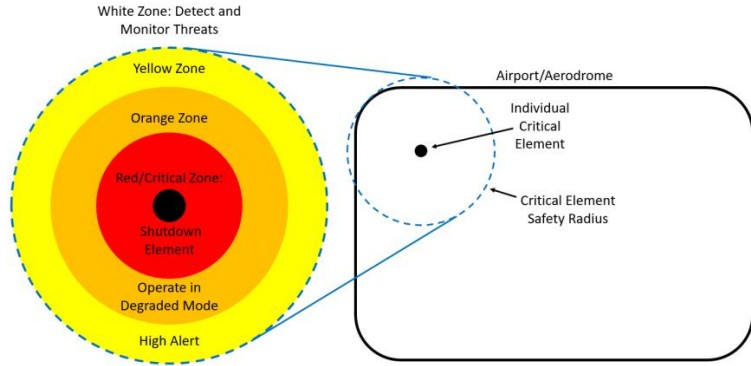

**Figure 2.** Alert levels and operational modes of a critical asset.

Note that the classification of the alert levels is strictly related to some configuration parameters, i.e., the sizes of the alert zones (yellow, orange, and red) and the functions available in the degraded operational mode. Such parameters are selected during the strategic phase of the ASPRID process model and depend on a number of features of the asset, such as its criticality to airport operations, its relationships with other assets, whether it is manned or not, whether the airport contains a redundant element or not, the presence of people around, etc. Moreover, this work assumes that the safety bubble and the alert levels of each asset are stationary and do not change with respect to the features of the intruder drone.

Figure 3 reports a tabular view of ASPRID's operational procedure, which is based on the aforementioned alert classification. The procedure addresses the actions to be performed by ASPRID operators and controllers (i.e., ATCOs), including tower controllers, ground controllers, and controllers for the terminal maneuvering area (TMA). These actions are triggered by specific conditions, which are mainly related to the alert levels in Figure 2. Only one condition is associated with the availability of a C-UAS action to neutralize an intruder drone: in this case, the ASPRID operator proceeds to issue the neutralization command only if authorized by the ATCO, who is responsible for checking that there are no moving aircraft near the target drone in order to avoid collateral damages. These guidelines are graphically specified in Figures 4 and 5, which, respectively, illustrate the BPMN process diagrams of the main process and of the infringement management subprocess within the ASPRID operational procedure. Note that, in Figure 4, the C-UAS management subprocess also includes LEAs; in Figure 5, the controller swimlane refers to ATCOs. Moreover, we assume that the main BPMN process (Figure 4) implies periodic detection and tracking for the ASPRID system; thus, the process starts on recurring moments in time by means of a timer start event.

| Triggering Condition | ASPRID Operator Action | Controller Action |
|---|---|---|
| White infringement of unauthorized drone | ① Monitor the object (periodic monitoring) <br> ② Keep the controller updated about the evolution of object's trajectory (significant updates) | No actions |
| Yellow infringement of unauthorized drone | ① Inform the controller <br> ② Monitor the object (continuous monitoring) <br> ③ Keep the controller updated about drone's trajectory and features, e.g., speed, model, etc. (continuous update) | ① Move/redirect aircraft to keep them far from the infringed zone (e.g., use another runway for taxiing or take-off) |
| Orange infringement of unauthorized drone | As above | ① Stop aircraft in nearby zones |
| Red infringement of unauthorized drone | As above | ① Stop all airport operations |
| Available countermeasure to neutralize an unauthorized drone | ① Ask the controller to confirm the neutralization action <br> ② If the controller confirms the neutralization action, issue neutralization command <br> ③ Inform the controller about the neutralization result <br> ④ Repeat in case of failed neutralization | ① Check that there are no moving aircraft close to the neutralization area (close to the drone) <br> ② Inform ASPRID operator about the previous check |

**Figure 3.** Tabular view of ASPRID operational procedure.

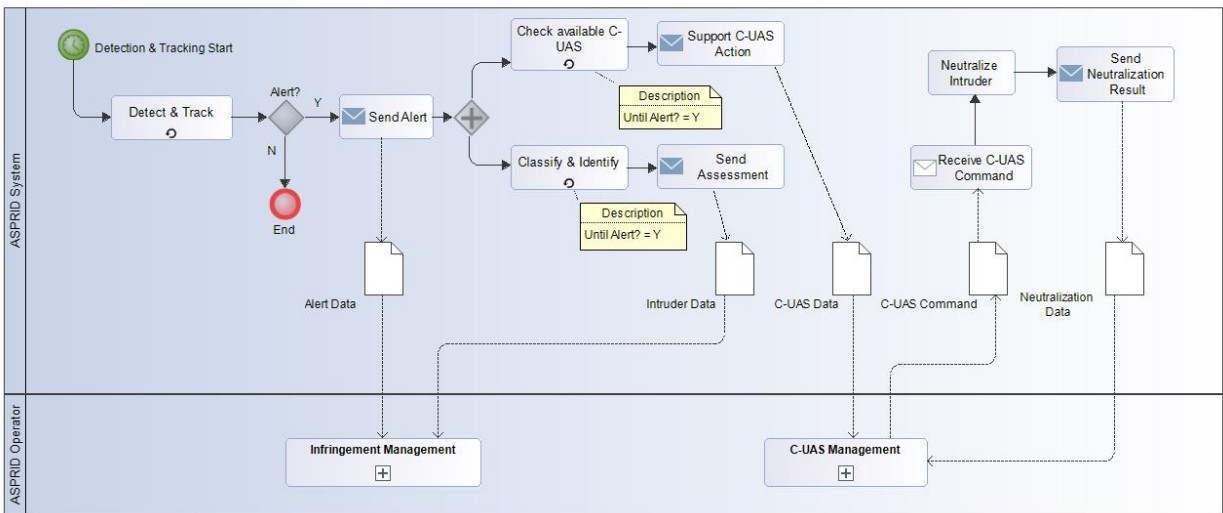

**Figure 4.** BPMN process diagram of ASPRID operational procedure: main process.

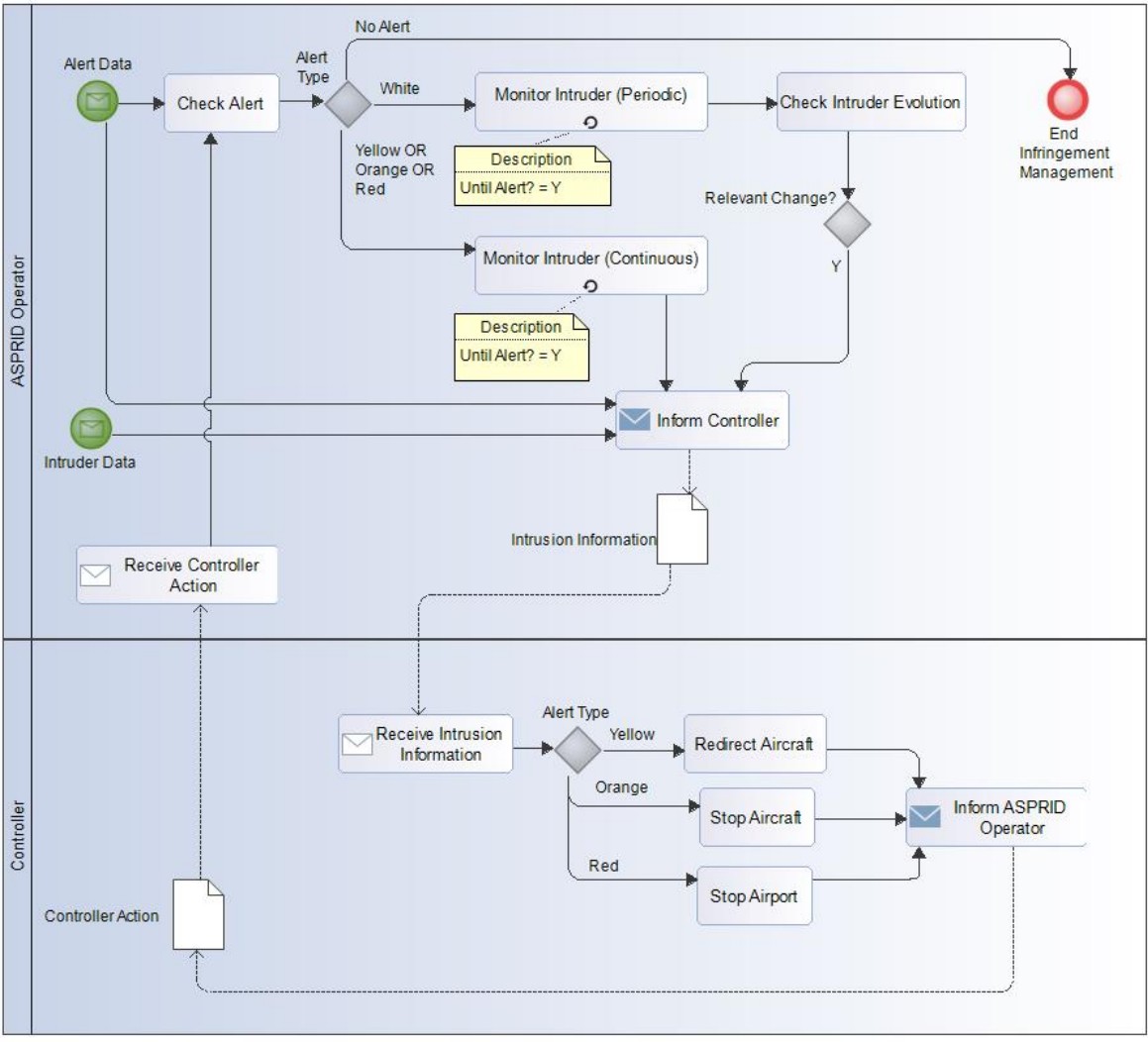

**Figure 5.** BPMN process diagram of ASPRID operational procedure: subprocess for infringement management.

The designed procedure stops airport operations only if a red-zone infringement occurs. Moreover, the procedure also implicitly considers aircraft restoring: in case of a downgrade of the alert level (e.g., in case of an unauthorized drone's transition from orange to yellow infringement or from red to orange infringement), the controller performs the specific actions for the new alert level, which possibly implies a restoring of some aircraft previously stopped, according to their position with respect to the infringement zone. Instead, the procedure does not currently consider the restoration times due to the possible removal of the neutralized drone within the airport's perimeter.

## 4. Evaluation Methodology

This section describes the methodology for the resilience-driven evaluation of AS-PRID's operational concept. It discusses the general approach and the simulation strategy.

### 4.1. Approach

A specific evaluation methodology is applied to assess the effectiveness of the AS-PRID operational concept for the protection of airport operations against drone intrusions. Considering the resilience-driven nature of the concept, such effectiveness is intended as a positive impact with regard to the availability of the DIMS and is demonstrated by assessing the main performance benefits related to the increase in the airport's resilience to drone intrusions by means of the DIMS itself.

A concrete case study is arranged in order to provide tangible results about the performance benefits. In detail, the Milan Malpensa Airport is chosen as the reference airport for the case study to evaluate the proposed operational concept. The ICAO (International Civil Aviation Organization) airport code is LIMC. This is a proper reference airport for this evaluation because it is of medium complexity in the context of European airports. The airport is located 49 km from central Milan and has two passenger terminals as well as a dedicated cargo terminal. It presents two runways in a parallel configuration with various taxiways connecting these with the aforementioned terminals and other airside areas, as shown in Figure 6.

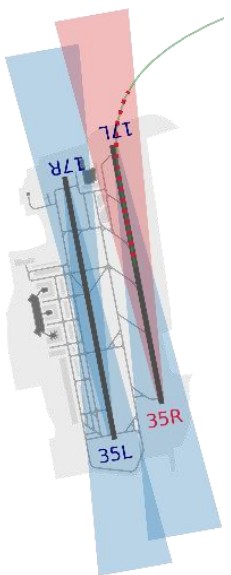

**Figure 6.** Runways and departure paths of Milan Malpensa Airport (LIMC).

The evaluation methodology combines: (i) a quantitative approach, based on real-time simulations; (ii) a qualitative approach, applying a risk analysis based on the achieved simulation results. The former approach involves the implementation of a proof-of-concept of the operational concept in a laboratory environment. Instead, the latter approach employs event trees.

Reference [14] provides a detailed description of the risk assessment of drone intrusions in airports, including event trees. The generic structure of such event trees is shown in Figure 7a, wherein the probability of a branch BProba is computed by using:

- Pts, the probability of the threat scenario;
- Pd, the probability of efficient drone detection;
- Pa, the probability of efficient drone assessment (i.e., identification);
- Pm, the probability of efficient drone mitigation (i.e., neutralization).

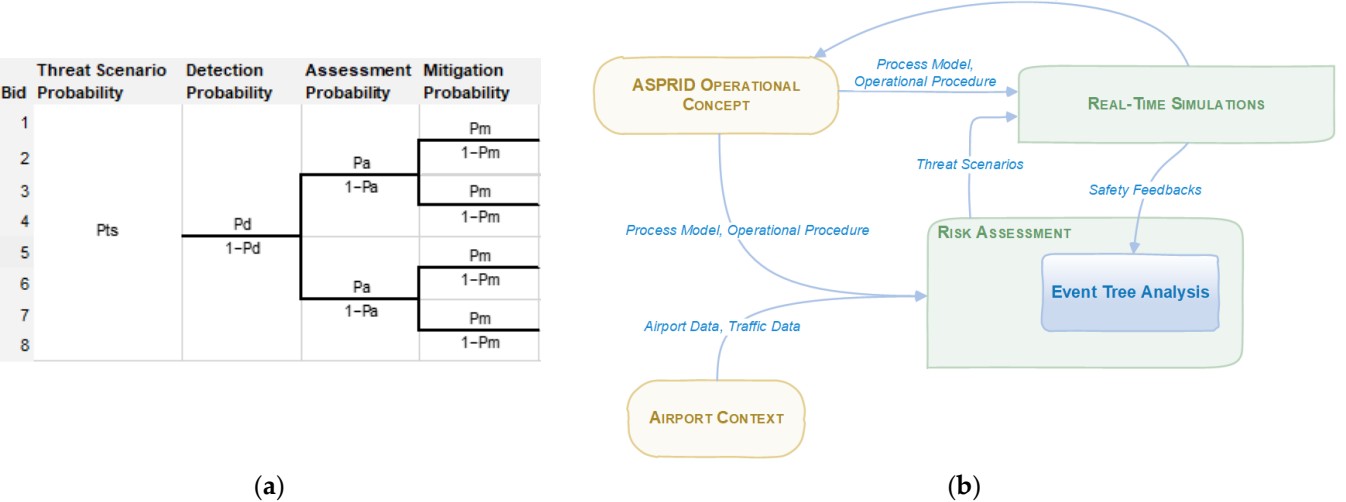

(**a**)     (**b**)

**Figure 7.** Evaluation approach. (**a**) Event tree for the assessment of threat scenarios [14]. (**b**) Evaluation workflow.

Instead, the proposed evaluation exploits real-time simulations to: (i) estimate resilience metrics and provide requirements for the implementation of the ASPRID system (C-UAS requirements); (ii) provide feedback to the risk assessment (ETA) in regard to safety aspects. The evaluation workflow is shown in Figure 7b. The following subsections provide further details about the approach for real-time simulations.

*4.2. Real-Time Simulations*

This section describes the simulation approach for the proposed evaluation. In detail, the approach uses real-time simulations, which means that some entities (actors or technological blocks) of the reference system are simulated by means of computer models, whereas other entities are played by human operators, requiring real-time constraints for simulation [30]. This approach is useful for evaluations that consider the behavior of human operators (e.g., pilots and controllers) and to validate the main functionalities of the reference system.

Moreover, given the functional scope of the ASPRID operational concept, the real-time simulations are intended as functional simulations in order to simulate the functional behavior as addressed by the specifications of the process model and the operational procedure. Thus, simulations represent "what" the ASPRID system shall do, not "how", by modeling its functional behavior. This allows for verifying the logical correctness of the concept and measuring its resilience-related benefits and impacts coherently from a functional viewpoint.

The following subsections point out: (i) the simulation platform; (ii) the simulation scenarios, including the operational environment and the threat scenarios; (iii) the performance metrics.

4.2.1. Simulation Platform

The simulation platform is a set of interconnected tools and apps to simulate both the airport environment and the ASPRID operational concept. The platform is able to replicate the expected behavior of the ASPRID system, providing its functional simulation. Thus, the platform represents a software proof-of-concept of the operational concept in order to collect quantitative measures about its functional behavior.

Figure 8 illustrates the architecture of the simulation platform, which is composed of the following tools and apps [31]:

- ASPRID server—This is a backend that simulates the ASPRID system. It receives the configuration of the ASPRID system, the drone tracks, the traffic tracks, and the countermeasure commands. It estimates the state of the airport related to the threat, delivering the outputs of the ASPRID system, i.e., the information about the detection and tracking, identification, and neutralization of the intruder. It was developed in TypeScript and it runs on the Node.js runtime environment.
- ASPRID configurator—This is the tool in charge of the configuration of the ASPRID system. It allows: (i) the tagging of the critical assets in the available airport and specification of the related alert zones; (ii) the locating of the sensing and countermeasure units and specification of their characteristics. It has been developed in TypeScript.
- eArts (en-route aerodrome real-time simulator)—This is a real-time simulator of aerodrome, en-route, and approach operations [32]. It can reproduce the controller working positions (CWPs) for the different roles of ATCOs and for aircraft pseudo-pilots (Figure 9). It was developed in the JAVA and Kotlin programming languages. Based on the outcomes of the ASPRID server, eArts CWPs are used to show: aircraft and drone tracks on the map; drone identification data; and alert information.
- ASPRID web app—This is a frontend for the interaction with different classes of users, such as (Figure 10): ASPRID's operator; a human viewer (described in Section 4.2.3); a drone operator (i.e., the user who selects the trajectory and the features of the intruder drone); an LEA operator; etc. The ASPRID operator uses this app to issue neutralization commands within a simulation run. The web app was developed with the Flutter/Dart language. A demo web app is publicly available to replay some simulation exercises [33].

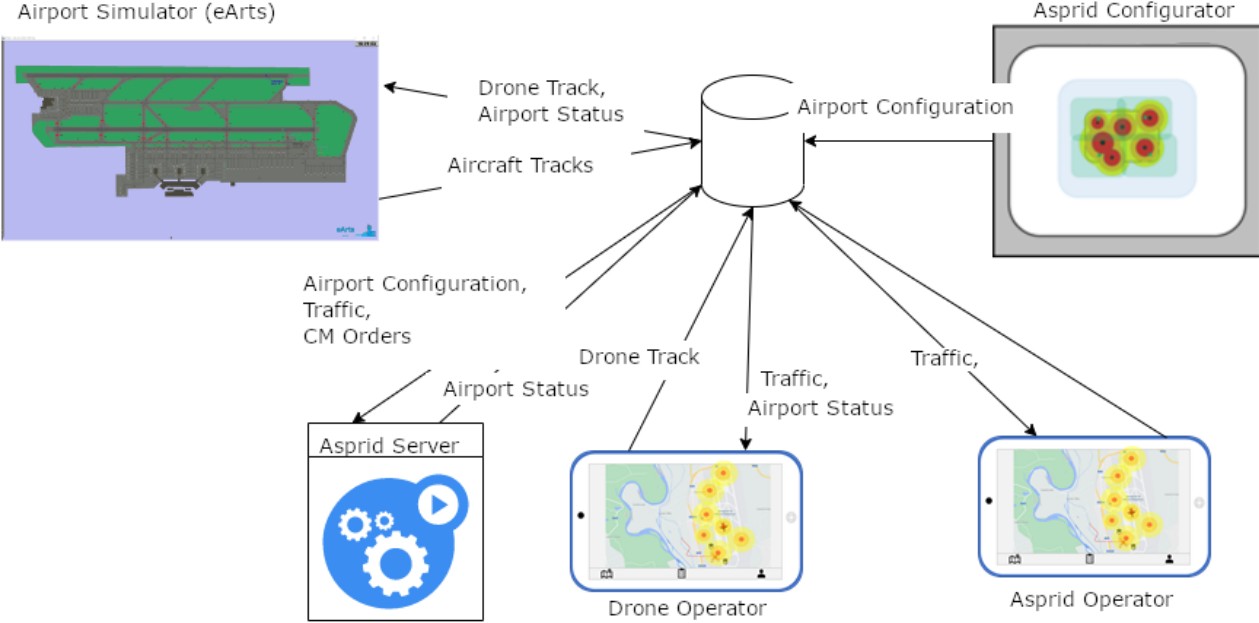

**Figure 8.** Architecture of the simulation platform.

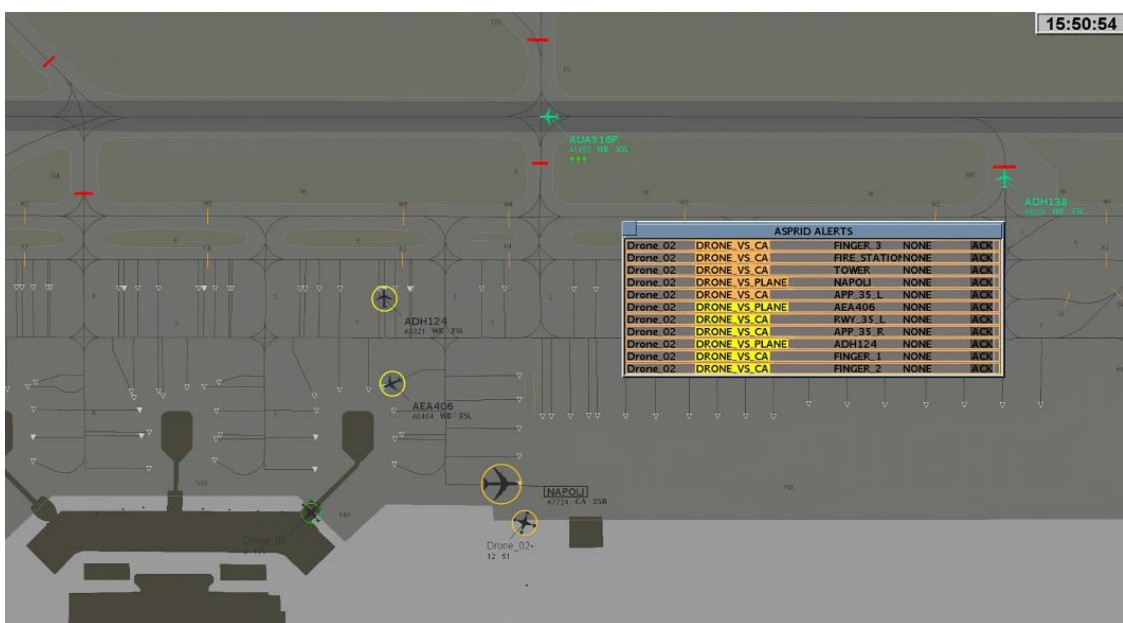

**Figure 9.** eArts simulator.

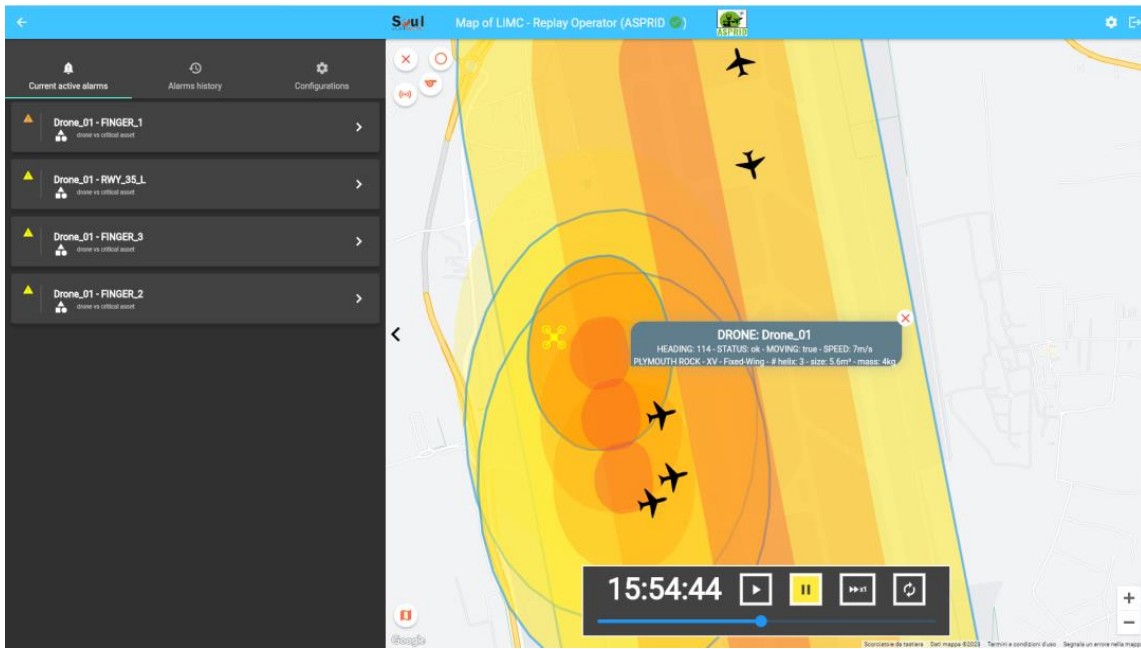

**Figure 10.** ASPRID web app.

Lastly, a specific ASPRID postprocessor collects the logs of the simulation runs and evaluates the performance metrics (described in Section 4.2.5). Due to the exploratory research carried out in the ASPRID project, the proposed evaluation does not consider the sensitivity of the results, even if the results should be able to be directly extended to all the scenarios (e.g., other airports) with the same traffic/operational environment.

### 4.2.2. Simulation Scenarios

Simulation scenarios are the representations of the environment and of the behaviors that are used for the real-time simulation of the proposed operational concept. The adopted simulation approach entails the following classes of simulation scenarios:

- The solution-scenario class, which is composed of simulation scenarios reproducing a given environment with the implementation of the operational improvement (i.e., the proposed operational concept) that is the subject of evaluation;
- The reference-scenario class, which is composed of simulation scenarios reproducing the same environment as a solution scenario without the implementation of the operational improvement that is the subject of evaluation.

Both classes are referred to within LIMC airport, coherently with the stated case study. Moreover, each simulation scenario is configured by defining its following components: (i) an airport environment, which in turn consists of the traffic environment and the operational environment for the selected airport; (ii) a threat scenario. The remainder of this subsection describes the traffic environment, whereas the operational environment and the threat scenarios are described, respectively, in Sections 4.2.3 and 4.2.4.

The traffic environment reproduces the air traffic distribution within the simulation scenario. All the solution scenarios and reference scenarios share the same traffic environment, which is built upon a real traffic distribution from a representative day at LIMC airport. A medium-traffic volume is used, which is between 50% and 75% of the high-traffic volume, and it includes only departing flights. In detail, all the simulation runs adopt the real planned departure traffic of LIMC for the day of 8 February 2019 at 15:50 GMT. The average time window used in the simulations is 40 min. The choice of using only departing flights is strictly related to the human team involved in the simulation runs, as further explained in Section 4.2.3. No other off-nominal events (in addition to drone intrusions) are included within the traffic environment.

### 4.2.3. Operational Environment

The operational environment reproduces the set of airport capabilities and procedures, including those related to counter-drone management operations. In detail, the following entities make up the simulation of the operational environment: (i) counter-drone sensing; (ii) counter-drone alerting; (iii) counter-drone neutralizing; (iv) human operators and counter-drone operational procedure.

Counter-drone sensing entities simulate the detection and tracking capabilities of the airport with respect to drone intrusions. Each sensor is modeled by means of the following parameters: spatial coverage, which is the geographic area covered by the sensor for the drone sensing; efficiency, which is the probability of successful sensing at each (simulated) time instant in case the drone is inside the covered area of the sensor.

For reference scenarios, counter-drone sensing consists only of human sightings, which are simulated by means of "artificial" sensors, which do not represent real physical sensors, but simulate the sighting of drones by humans. Human sightings are modeled only by stationary sensors and by properly configuring their parameters. The following classes of sensors are used for human sightings in reference scenarios:

- Artificial airport sensors—They are displaced within the airport perimeter. They also include an artificial tower sensor, related to ATCOs in the tower.
- Artificial external sensors—They are displaced outside the airport perimeter. They simulate the human sighting of drones from the outside (e.g., from aircraft pilots).

For solution scenarios, the following classes of sensors are used to simulate drone sensing technologies according to the ASPRID operational concept:

- Radar sensors—They simulate the detection-and-tracking capabilities by means of radars. They are able to detect and track the radar signature of an object, but they are not able to identify its nature. Thus, radars detect and track unknown objects, but they cannot confirm whether the object is a real unauthorized drone.
- Identification sensors—They simulate the detection-and-tracking capabilities by means of specific sensors (e.g., optical sensors, acoustic sensors, infrared sensors, etc.), which may also provide identification. They are able to: detect and track an object; confirm whether it is an unauthorized drone; recognize the model; provide an estimation

of speed, mass, and payload. This class of sensors is divided into the following subclasses: (i) airport identification sensors, which represent identification sensors displaced within the airport perimeter; (ii) external identification sensors, which represent identification sensors displaced outside the airport perimeter, simulating the identification of drones by means of external support, such as LEAs.

Table 1 reports the parameter values of the counter-drone sensing entities. For reference scenarios, the configuration of the spatial coverages and of the efficiency is based on qualitative criteria related to the line of sight and the density of people on the ground. For solution scenarios, the parameter values reflect the technological features of the sensors according to a specific technological analysis carried out within the ASPRID project.

**Table 1.** Parameter values of the counter-drone sensing entities.

| Simulation Scenarios | Sensor Class | Spatial Coverage (m) | Efficiency |
| --- | --- | --- | --- |
| Reference | Artificial airport sensors (tower) | 500 | 50% |
| | Artificial airport sensors (other) | 400 | 50% |
| | Artificial external sensors | 600 | 50% |
| Solution | Radar sensors | 3500 | 25% |
| | Identification sensors | 1000 | 75% |

Figures 11 and 12, respectively, show the displacement of counter-drone sensors within reference scenarios and solution scenarios. For reference scenarios, the displacement of artificial airport sensors qualitatively considers the density of human actors on the ground during the intrusion, favoring sensing near critical sites such as the taxiways, the parking area, etc. For solution scenarios, displacement is a design choice for the operational concept within the case study.

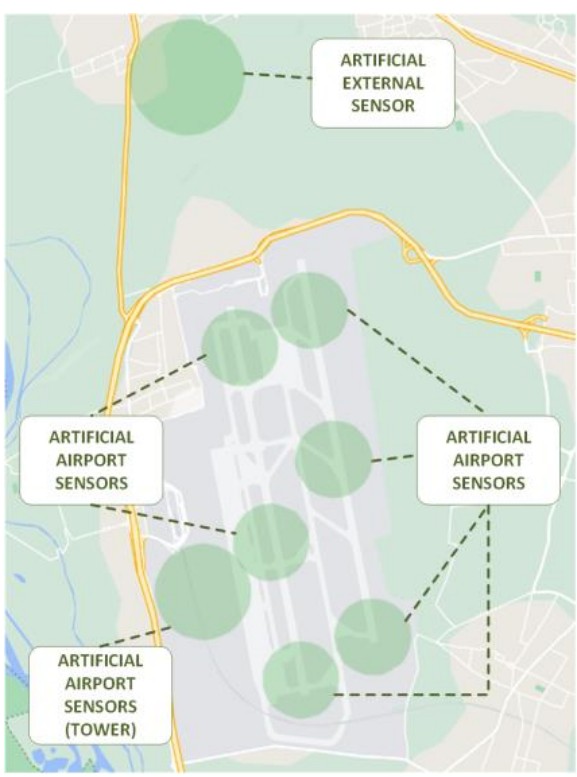

**Figure 11.** Displacement of counter-drone sensors for reference scenarios.

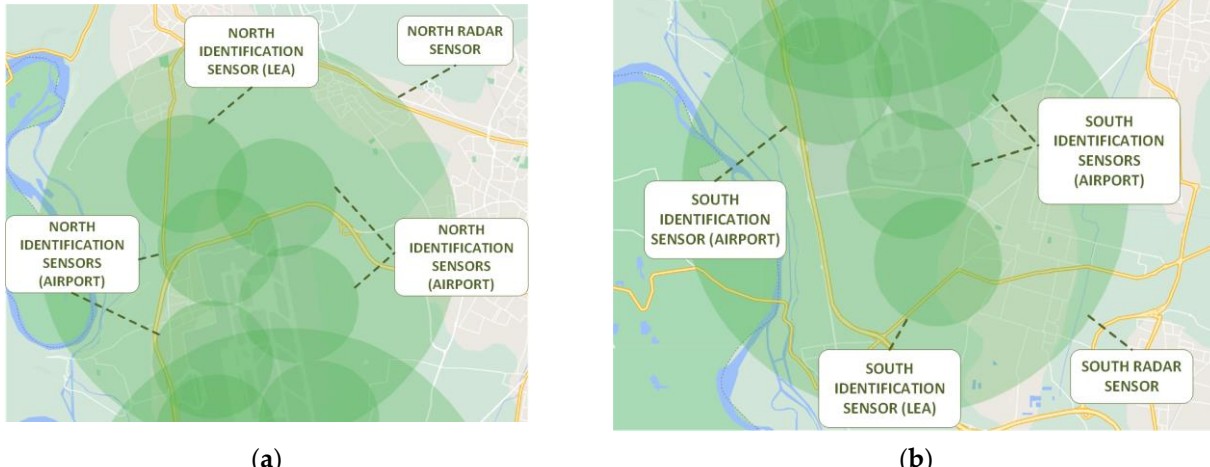

(**a**)                                                    (**b**)

**Figure 12.** Displacement of counter-drone sensors for solution scenarios. (**a**) North part of LIMC. (**b**) South part of LIMC.

Counter-drone alerting entities simulate the alerting capabilities of the airport with respect to drone intrusions in the vicinity of a critical asset. These entities are present only in solution scenarios, and their configuration applies the following rules according to the ASPRID operational concept:

1. Each asset is associated with three alerting zones, i.e., yellow, orange, and red;
2. The risk levels of the alert are in ascending order from yellow to orange and then to red, and consider the distance of the drone from the asset;
3. The geometrical shape of an alerting zone may be any closed shape.

The covered stationary assets of LIMC are: runway, 35L; runway, 35R; approach area of runway, 35L (APP 35L); approach area of runway, 35R (APP 35R); departure area of runway, 35L (DEP 35L); departure area of runway, 35R (DEP 35R); tower; finger 1; finger 2; finger 3; fire station. Figure 13 highlights the stationary critical assets and illustrates the displacement of their alerting zones. In regard to the aircraft, the following extensions of the alerting zones are used: 700 m for the yellow zone; 100 m for the orange zone; 50 m for the red zone. With regard to the stationary critical assets, a size of 600 m is used for the yellow zones of all the assets. Different values are used for the orange and red zones, related to the effective dimension of the asset.

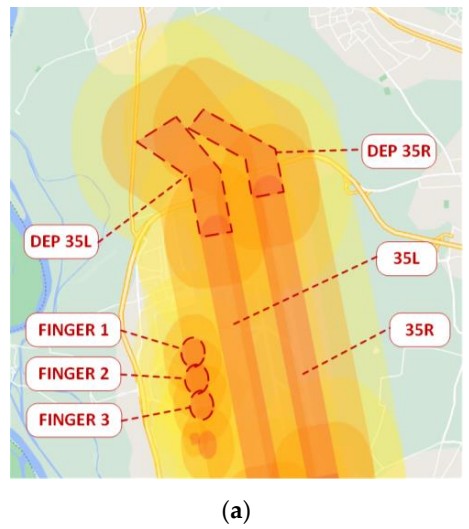     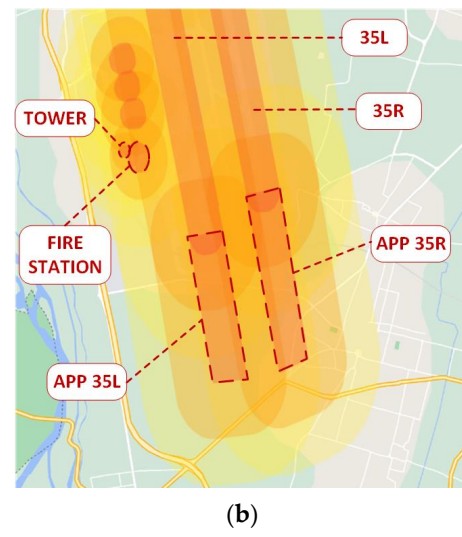

(**a**)                                                    (**b**)

**Figure 13.** Displacement of counter-drone alerting zones of stationary assets for solution scenarios. (**a**) North part of LIMC. (**b**) South part of LIMC.

Counter-drone neutralization entities simulate the neutralization technologies and the related neutralizers or countermeasures. These entities are present only in solution scenarios. For the purposes of the proposed evaluation, jamming and net protection (i.e., physical interception) are simulated. Jammers, especially, are deemed more common with respect to spoofing; thus, they are part of the operational environment. Neutralizers are modeled by means of the following parameters: latitude and longitude of the position of the neutralizer; spatial coverage, i.e., the radius of the zone covered for the neutralization action; reaction time, i.e., the latency between the issuing of the neutralization command from the ASPRID operator and the execution of the neutralization action; success probability, referring to a single neutralization attempt. The following neutralizers are used:

- Airport jammers (internal to the airport);
- LEA jammers (external to the airport);
- Airport net.

Table 2 reports the parameter values of the counter-drone neutralization entities, which are selected according to a specific technological analysis carried out within the ASPRID project. Figure 14 shows the displacement of neutralizers that is adopted for the evaluation within the case study.

**Table 2.** Parameter values of the counter-drone sensing entities.

| Neutralizer | Spatial Coverage (m) | Reaction Time (s) | Success Probability |
|---|---|---|---|
| Airport jammer | 1100 | 1 | 65% |
| LEA jammer | 1300 | 30 | 65% |
| Airport net | 100 | 2 | 50% |

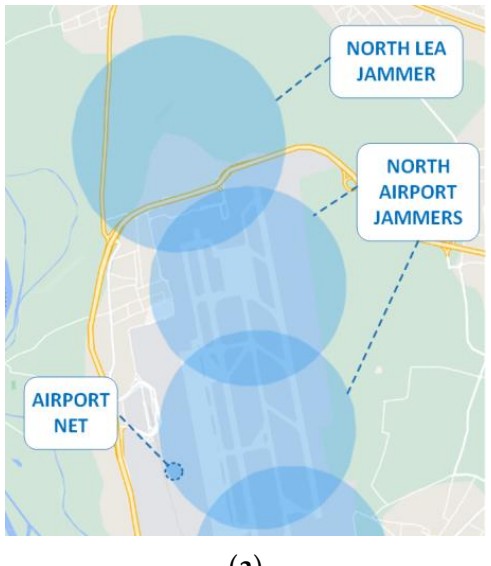

(**a**)

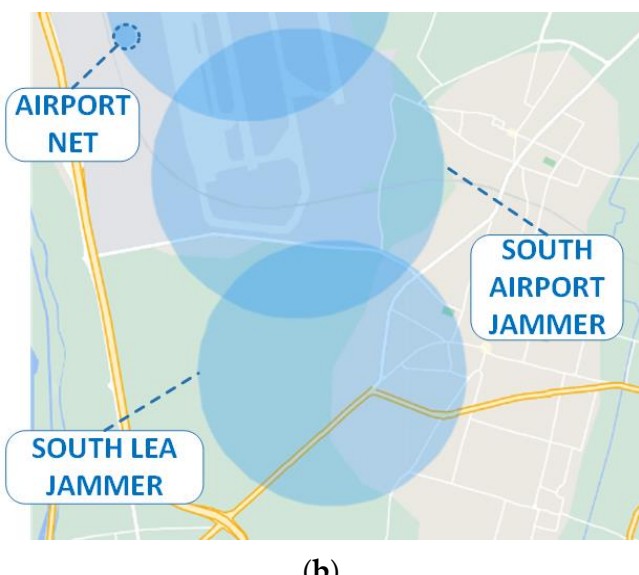

(**b**)

**Figure 14.** Displacement of counter-drone neutralization entities for solution scenarios. (**a**) North part of LIMC. (**b**) South part of LIMC.

The counter-drone operational procedure represents the procedure to manage drone intrusions within the operational environment. For reference scenarios, the following operators are involved: (i) one person plays the "controller" role, which includes both the ground controller and the tower controller; (ii) one person plays the "human viewer" role for the sighting of unauthorized drones and for informing the controller in case of sightings. The following procedure is adopted:

1. In case of a drone sighting by a human viewer, this viewer informs the controller;

2. The controller stops all ongoing operations and does not allow new aircraft to taxi from those available at the gates.

Considering the average time window of the simulations (40 min), airport operations are never resumed after an unauthorized drone sighting. Moreover, in the case of a collision between the drone and a critical airport asset, the simulation run is stopped.

Instead, for solution scenarios, the following operators are involved: (i) one person plays the controller role, which includes both the ground controller and the tower controller; (ii) one person plays the "ASPRID operator" role. The counter-drone operational procedure is the one representing the ASPRID operational procedure, described in Section 3.2.

No pseudo-pilot is involved in the simulations of either reference scenarios or solution scenarios. Thus, it is assumed that the aircraft always executes the controller's instructions.

### 4.2.4. Threat Scenarios

Threat scenarios or hazard scenarios are descriptions of how a threat or hazard might materialize and represent a logical sequence from a hazard to its consequence [10]. A threat scenario specifies the chain of events or occurrences that take place, starting with one or more threats and ending with the consequences of an incident [34]. In this work, threats are unauthorized drone intrusions in the airport, and threat scenarios specify the chain of events or occurrences that:

- Start with a drone intrusion;
- Affect the operations of the airport and of its DIMS (if any);
- End with the consequences of an incident (i.e., collision with an airport's asset) or with a safe termination of the intrusion (e.g., the exit of the drone from the airport and its surroundings, neutralization of the drone, etc.).

For the purposes of the proposed resilience evaluation, an extended risk assessment process has to be adopted with respect to threat scenarios, in order to include both the resilience-driven and the simulation-based perspectives. Indeed, all the threat scenarios should be simulated within both reference scenarios and solution scenarios to: (i) quantify the level of risk for each threat scenario; (ii) quantify the level of the airport's resilience for each threat scenario; (iii) compare the results between reference and solution scenarios to provide a quantitative evaluation of the expected performance of the operational concept.

In general, threat scenarios should be assessed by considering the possible physical interference of an intruder drone with respect to a given airport's assets. In principle, such behavior may be traced to two main classes of drone interference: (i) fly-by, which refers to the proximal presence of a drone with respect to an asset or an airspace; (ii) collision, which refers to the crash of the drone with an asset, usually implying damage to the asset according to the kinetic energy of the drone. Several control variables may influence the evolution of a threat scenario and the outcomes of the evaluation, such as:

- The features of the intruder, e.g., in terms of trajectory, speed (also including constant-speed intrusion or variable-speed intrusion), drone type and related mass, etc.;
- The effectiveness of the sensing technologies of the DIMS, e.g., in terms of space coverage, accuracy, acquisition rate, success probability, dependency on meteorological conditions, etc.;
- The effectiveness of the neutralization technologies of the DIMS, e.g., in terms of technology (e.g., jamming, protection net, etc.), spatial coverage, latency, success probability, stationary/mobile type, dependency on meteorological conditions, collateral damages, etc.;
- The spatial displacement of the sensor and neutralization components;
- The type of airport traffic.

Thus, the evaluation of an operational concept for the DIMS is generally a complex problem due to the large number of threat scenarios, control variables, and their values. However, considering the exploratory nature of ASPRID research, the proposed evaluation does not assess the influence of the aforementioned variables on the outcomes, except for

the bidimensional intrusion trajectory. Indeed, this is deemed an essential variable since it differentiates the main plausible classes of threat scenarios based on the airport's asset(s) that are affected by the intrusion trajectory, including standard instrument departure routes (SIDs) and standard arrival routes (STARs). The speed of the drones is simulated as a constant in the range of 5 ÷ 10 m/s, which reflects the most common speed of commercial drones. In regard to the intrusion's altitude, it is assumed to be below a threshold that would allow the intrusion to interfere with airport operations. Such constraints are related to the simulation tools, whereas the ASPRID operational concept is basically a 3D solution.

Table 3 lists the threat scenario classes that have been identified for LIMC and the instances of the classes (with the related figures) that have been simulated for the evaluation.

**Table 3.** Threat scenario classes and related instances for the proposed evaluation.

| Threat Scenario Class | Description | Instances |
|---|---|---|
| Threat on 35L | This class includes intrusion trajectories representing direct threats to the runway 35L. | Drone 01—It affects the north part of 35L (Figure 15)<br>Drone 02—It affects the middle part of 35L (Figure 15)<br>Drone 03—It affects the south part of 35L (Figure 15) |
| Threat on 35R | This class includes intrusion trajectories representing direct threats to the runway 35R. | Drone 01—It affects the north part of 35R (Figure 15)<br>Drone 02—It affects the middle part of 35R Figure 15)<br>Drone 03—It affects the south part of 35R (Figure 15) |
| Threat parallel to RWYs | This class includes intrusion trajectories representing direct threats running parallel to the runways. | Drone 01 and Drone 04—They affect 35L (Figure 16)<br><br>Drone 02 and Drone 03—They affect 35R (Figure 16) |
| Threat on fingers | This class includes intrusion trajectories representing direct threats to the fingers. | Drone 01 (Figure 17) |
| Threat on tower | This class includes intrusion trajectories representing direct threats to the tower. | Drone 01 (Figure 17) |
| Threat on routes | This class includes intrusion trajectories representing direct threats to the routes, i.e., for the SIDs and STARs of the airport. | Drone 01—It affects the SIDs, without entering the airport perimeter (Figure 18)<br>Drone 02—It affects the STARs, without entering the airport perimeter (Figure 18)<br>Drone 03—It affects the SIDs and then enters the airport perimeter (Figure 18)<br>Drone 04—It affects the STARs and then enters the airport perimeter (Figure 18) |

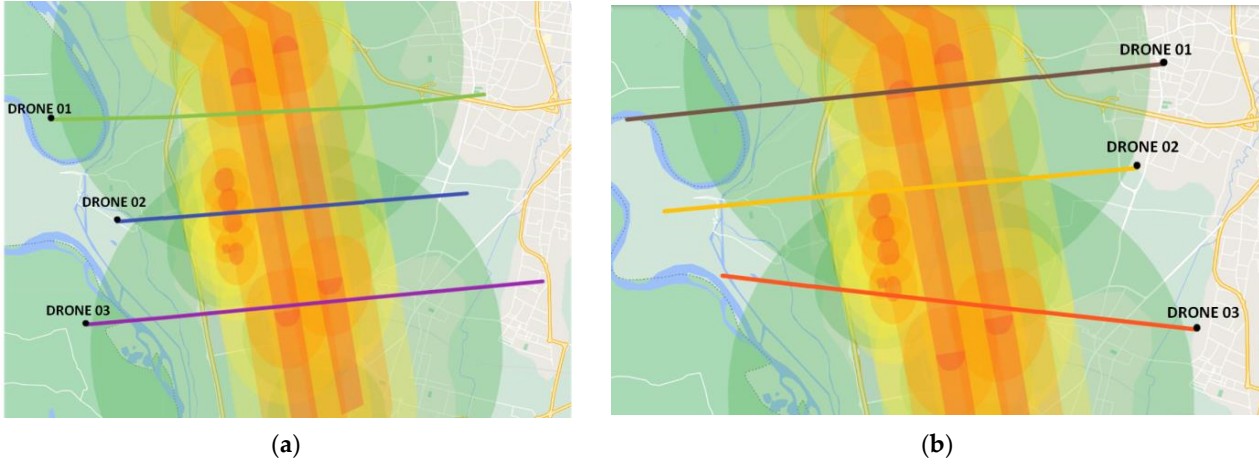

(**a**)          (**b**)

**Figure 15.** Instances of threat scenario classes. (**a**) "Threat on 35L". (**b**) "Threat on 35R".

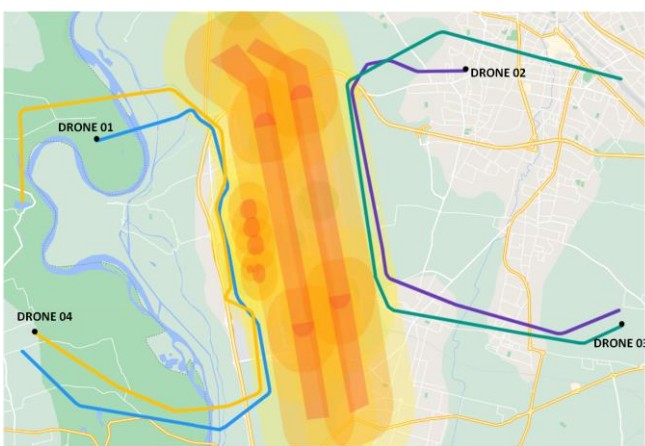

**Figure 16.** Instances of threat scenario class "Threat parallel to RWYs".

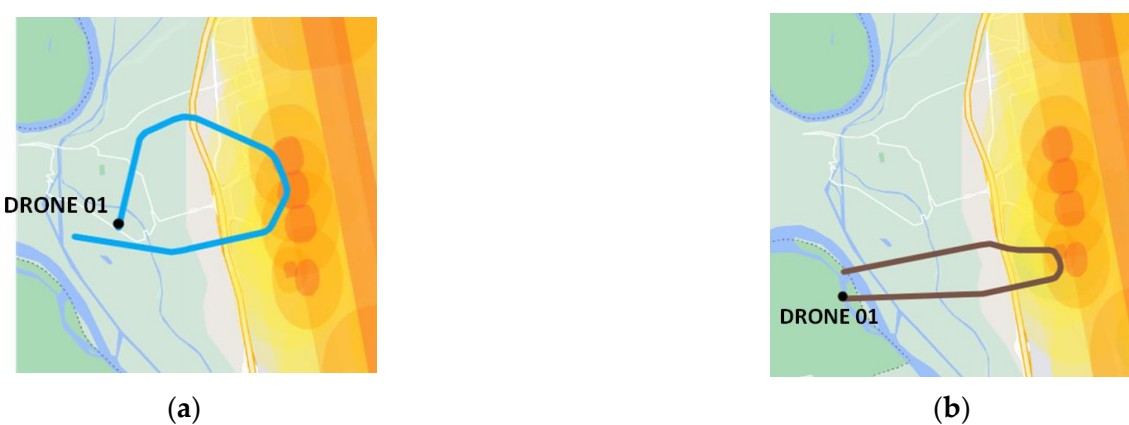

(**a**)                                    (**b**)

**Figure 17.** Instances of threat scenario classes. (**a**) "Threat on fingers". (**b**) "Threat on tower".

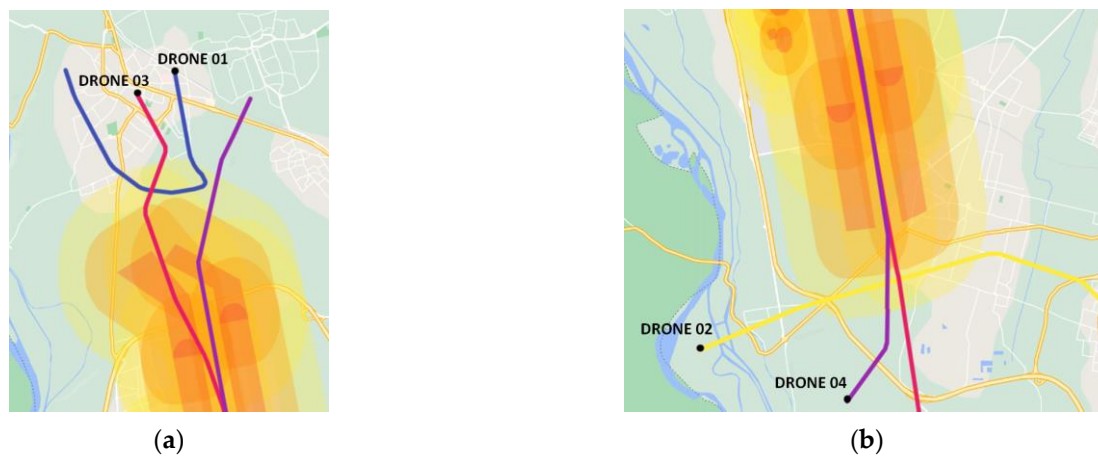

(**a**)                                    (**b**)

**Figure 18.** Instances of threat scenario classes. (**a**) "Threat on routes (SIDs)". (**b**) "Threat on routes (STARs)".

### 4.2.5. Performance Metrics

The basic idea of the proposed simulation-based evaluation consists of a quantitative comparison between the simulation results of reference scenarios and solution scenarios. This requires the setup of a specific performance framework, i.e., a proper set of performance metrics to allow the measurement of the effectiveness of the operational concept from a resilience-driven perspective.

The main reference for the definition of such a performance framework is represented by the SESAR performance framework [35], which reports: (i) the performance management process for measuring the performance and tracking the status of SESAR solutions with regard to performance targets; (ii) a set of key performance areas (KPAs), key performance indicators (KPIs), and performance indicators (PI). In addition, the SESAR performance framework specifies some focus areas within each KPA that are typically related to some identified performance issues of ATM. In detail, a resilience focus area is addressed within the capacity KPA in order to measure the ability [35]:

- To deliver performance goals during abnormal operating conditions by reducing impact and decreasing time to recover;
- To withstand and recover from planned and unplanned events and conditions that cause a loss of nominal capacity.

In this focus area, the resilience of an ATM system is influenced by [35]: (i) its anticipation and handling in regard to degraded conditions; (ii) its recovery from degraded to normal conditions. Planned and unplanned events and conditions include: weather, such as thunderstorms, strong winds, freezing conditions, and low visibility conditions; infrastructure degradation, such as technical failures, strikes, accidents, and runway maintenance.

However, the SESAR performance framework does not explicitly consider airports' counter-drone operations. Thus, within the proposed simulation-based evaluation, some specific performance metrics are defined to measure: (i) the anticipation and handling of airports in regard to degraded conditions due to drone intrusions; (ii) the recovery of airports from degraded to normal conditions in the case of drone intrusions. For this purpose, the following sets of metrics are introduced, based on the data provided by the simulation tools (as described in Section 4.2.1):

- Safety metrics (Table 4)—They measure the variation in safety margins for airport operations due to drone intrusions. They are defined in connection with the critical assets, the threat scenarios, and the operational procedure for drone intrusions. They basically consider the infringements of the intrusions with respect to the critical assets of the airports.
- Time performance metrics (Table 5)—They measure the variation in time performance for airport operations due to drone intrusions. They consider the punctuality of the airport traffic.
- Workload metrics (Table 6)—They measure the variation in human workload for airport operations due to drone intrusions. The human workload is related to controllers and is basically quantified in terms of the number of clearances.
- Capacity metrics (Table 7)—They measure the variation in airport capacity due to drone intrusions. Capacity is assessed by means of arrival and departure rates for each runway.

**Table 4.** Safety metrics.

| Safety Metric | Description |
|---|---|
| NYI | Number of yellow-zone infringements (NYI) of drone intrusions |
| NOI | Number of orange-zone infringements (NOI) of drone intrusions |
| NRI | Number of red-zone infringements (NRI) of drone intrusions |
| MTYI | Mean time of yellow-zone infringement (MTYI) of drone intrusions |
| MTOI | Mean time of orange-zone infringement (MTOI) of drone intrusions |
| MTRI | Mean time of red-zone infringement (MTRI) of drone intrusions |
| WTYI | Worst time of yellow-zone infringement (WTYI) of drone intrusions |
| WTOI | Worst time of orange-zone infringement (MTOI) of drone intrusions |
| WTRI | Worst time of red-zone infringement (MTRI) of drone intrusions |
| NYI-P | Number of yellow-zone infringements of drone intrusions against planes |
| NOI-P | Number of orange-zone infringements of drone intrusions against planes |
| NRI-P | Number of red-zone infringements of drone intrusions against planes |
| NYI-A | Number of yellow-zone infringements of drone intrusions against stationary assets |
| NOI-A | Number of orange-zone infringements of drone intrusions against stationary assets |
| NRI-A | Number of red-zone infringements of drone intrusions against stationary assets |

**Table 5.** Time performance metrics.

| Time Performance Metric | Description |
| --- | --- |
| NDAF | Number of delayed arrival flights (NDAF) with a delay greater than a given threshold |
| NDDF | Number of delayed departing flights (NDDF) with a delay greater than a given threshold |
| PDAF | Percentage of delayed arrival flights (PDAF) with a delay greater than a given threshold |
| PDDF | Percentage of delayed departing flights (PDDF) with a delay greater than a given threshold |
| MDAF | Mean delay of arrival flights (MDAF) with a delay greater than a given threshold |
| MDDF | Mean delay of departing flights (MDDF) with a delay greater than a given threshold |
| WDAF | Worst delay of arrival flights (WDAF) with a delay greater than a given threshold |
| WDDF | Worst delay of departing flights (WDDF) with a delay greater than a given threshold |

**Table 6.** Workload metrics.

| Workload Metric | Description |
| --- | --- |
| NCAF | Number of controller clearances for arrival flights (NCAF) |
| NCDF | Number of controller clearances for departing flights (NCDF) |
| NNCAF | Normalized number of controller clearances for arrival flights (NNCAF) |
| NNCDF | Normalized number of controller clearances for departing flights (NCDF) |

**Table 7.** Capacity metrics.

| Capacity Metric | Description |
| --- | --- |
| AR | Arrival rate (AR) per 15 min for a given runway |
| DR | Departure rate (AR) per 15 min for a given runway |

In addition to the aforementioned metric sets, some comprehensive performance degradation metrics are introduced to provide synthetic and quantitative indexes about the overall resilience, i.e., the performance degradation of airport operations due to drone intrusions. Every performance degradation metric consists of a classification scale with different degradation levels, each of which is associated with a triggering rule specifying the logical condition that underlies the corresponding level. The following are the proposed performance degradation metrics:

- Safety degradation metric (SDM, Table 8)—It includes four degradation levels (catastrophic, hazardous, major, minor, and negligible), which are related to the infringements of the assets by means of drone intrusions. The triggering rules in Table 8 do not consider: collisions between drones and assets, since red-zone infringements already trigger catastrophic (for aircraft) and hazardous (for stationary assets) degradations; durations and speed criticalities of the intruder trajectories in the infringement zones, since these are not used as control variables for the threat scenarios, as explained in Section 4.2.4.
- Time performance degradation metric (TDM, Table 9)—It includes three degradation levels (high, medium, and low), which are related to the mean delays for arrival and departure traffic.
- Workload degradation metric (WDM, Table 10)—It includes three degradation levels (high, medium, and low), which are related to the normalized numbers of controllers' clearances to manage arrival and departure flights.
- In the tables for the degradation metrics (Tables 8–10), the triggering rules are applied in an ordered way, starting from the ones applicable for greater degradation levels. For example, in Table 9 for TDM, the rule for the medium level is applicable if the rule for the high level is not triggered; the rule for the low level is applicable if the rules for the high and medium levels are not triggered.

The proposed metrics represent the possible indicators for a performance framework to evaluate the counter-drone behaviors of airports and their resilience against drone intrusions. Starting from these metrics, further specific indexes may be defined to evaluate additional resilience aspects (e.g., the percentage of diverted arrival traffic due to drone

intrusions) or additional performance dimensions, such as the one related to the effectiveness of the counter-drone technology (e.g., the success rate of the overall detection system, the success rate of the overall neutralization system, etc.).

**Table 8.** Safety degradation metric.

| Safety Degradation Level | Safety Degradation Value | Triggering Rule |
|:---:|:---:|:---:|
| Catastrophic | 4 | $NRI - P > 0$ |
| Hazardous | 3 | $(NOI - P > 0)$ OR $(NRI - A > 0)$ |
| Major | 2 | $(NYI - P > 0)$ OR $(NOI - A > 0)$ |
| Minor | 1 | $NYI - A > 0$ |
| Negligible | 0 | All other conditions |

**Table 9.** Time performance degradation metric.

| Time Performance Degradation Level | Time Performance Degradation Value | Triggering Rule |
|:---:|:---:|:---:|
| High | 2 | $(MDAF > 20\ min)$ OR $(MDDF > 20\ min)$ |
| Medium | 1 | $(MDAF \in [10\ min,\ 20\ min])$ OR $(MDDF \in [10\ min,\ 20\ min])$ |
| Low | 0 | $(MDAF < 20\ min)$ OR $(MDDF < 20\ min)$ |

**Table 10.** Workload degradation metric.

| Workload Degradation Level | Workload Degradation Value | Triggering Rule |
|:---:|:---:|:---:|
| High | 2 | $(NNCAF > 10)$ OR $(NNCDF > 10)$ |
| Medium | 1 | $(NNCAF \in [5, 10])$ OR $(NNCDF \in [5, 10])$ |
| Low | 0 | All other conditions |

## 5. Results

This section describes the evaluation results, highlighting both the simulation and risk assessment results.

### 5.1. Simulation Results

In line with the approach described in Section 4, each threat scenario was simulated in order to estimate the performance metrics for both the reference scenarios and solution scenarios. The tables in Appendix A (Tables A1–A5) report the detailed results for each metric set (safety, time performance, workload, and capacity). Considering the sensing configuration for reference scenarios, some of the designed threat scenarios ("threat parallel to runways—Drone 02 and Drone 03", "threat on routes—Drone 02 and Drone 04") were deemed non-significant for the simulations since they did not imply a drone sighting. Thus, such threat scenarios were not included in the evaluation.

The following considerations may be made relative to the performance analysis of the simulation results:

- Safety (Tables A1 and A2)—For the sake of brevity, only a subset of the performance metrics has been reported in the tables.

  a. Reference scenarios—All the threat scenarios lead to a catastrophic (4) value of safety degradation, as shown in Table A1. Several threat scenarios imply a collision between the intruder drone and a stationary asset, such as the scenarios "threat on 35L" and "threat on 35R".

  b. Solution scenarios—No threat scenarios imply collisions between drones and airport assets and no catastrophic (4) values are recorded for the safety degradation, as shown in Table A2. The only situation with a hazardous (3) SDM is represented by the scenario "threat on fingers—Drone 01", due to an orange zone

infringement with a plane. However, looking at the trajectory of the intruder in Figure 17, this infringement regards only stationary planes at the fingers. Thus, solution scenarios limit the reduction of safety margins.

- Time performance (Table A3)—With respect to the metrics in Table 5, two metrics are added in Table A3, which are: the number of departures, i.e., the number of flights available at the gates and ready for taxiing, and the number of takeoffs. Note that the number of departures is generally greater in solution scenarios since reference scenarios stop departure operations in the case of a drone sighting. Moreover, the number of takeoffs is always less than the number of departures, even in solution scenarios. This is because, in the case of drone neutralization, recovery is simulated by just restoring the previously stopped aircraft without approving new aircraft for taxiing from those available at the gates.

  a. Reference scenarios—Almost all the threat scenarios lead to a high (2) time performance degradation. Only the scenarios "threat on 35R—Drone 01" and "threat parallel to runways—Drone 04" lead to medium (1) and low (0) values for TDM. This is due to the specific trajectory of the drone intrusion in these scenarios, which is such that it postpones the drone sighting and implies fewer delays. This occurs even if a collision is recorded for the scenario "threat on 35R—Drone 01" considering that the collision occurs "late" in the simulation timeframe.

  b. Solution scenarios—Almost all the threat scenarios lead to a low (0) degradation of time performance, with the exception of the scenario "threat on routes—Drone 01". This scenario is especially challenging from the point of view of the operational procedure since the drone's trajectory (Figure 18) is such that no neutralization is possible. For the given scenarios and in the simulated timeframe (on average, 40 min), the ASPRID operational concept achieves a mean departing delay of about 400 s, which is less than a third of the mean departing delay of reference scenarios (about 1360 s). On the one hand, this gives a clear and quantitative idea of the time performance improvement in solution scenarios with respect to reference scenarios. On the other hand, this improvement is underestimated since several flights in reference scenarios would garner greater delays (or would even be canceled) with longer simulation timeframes.

- Workload (Table A4)—The number of departures is included also in the analysis of workload metrics, as it is for the time performance.

  a. Reference scenarios—Almost all the threat scenarios lead to a low (0) degradation of workload. This is coherent with the idea that, in reference scenarios, the operational procedure simply stops all airport operations in the case of drone sightings without providing a significant burden to controllers' workload. However, this is also due to the fact that for these scenarios, the simulation never restores airport operations after an unauthorized drone sighting; in the case of a collision between the drone and a critical airport asset, the simulation is stopped without considering restoration.

  b. Solution scenarios—Almost all the threat scenarios lead to a medium (1) degradation of workload. This is coherent with the greater awareness and the early management of drone intrusions and the new guidelines of the ASPRID operational procedure, which prescribe different controller actions according to the current alert level of the threat. However, only a slight workload increase is observed in solution scenarios. Indeed, the normalized number of clearances per departing flight is on average 5.14, which is only 0.3 greater with respect to reference scenarios (4.86).

- Capacity (Table A5)—The results for capacity provide an additional view with respect to time performance. First of all, the same observations may be made related to the number of flights released for departures. Moreover, a clear capacity improvement

may be observed. Indeed, solution scenarios show an average increment of about four flights/15 min for the departure rate with respect to reference scenarios.

By way of example, Figure 19 shows the performance results for the MDDF (belonging to the time performance set) and for the departure rate (belonging to the capacity set), also highlighting the mean values (red vertical lines) and the intervals for the standard deviation (yellow vertical lines). This figure gives a clear and quantitative idea of time performance and capacity improvements.

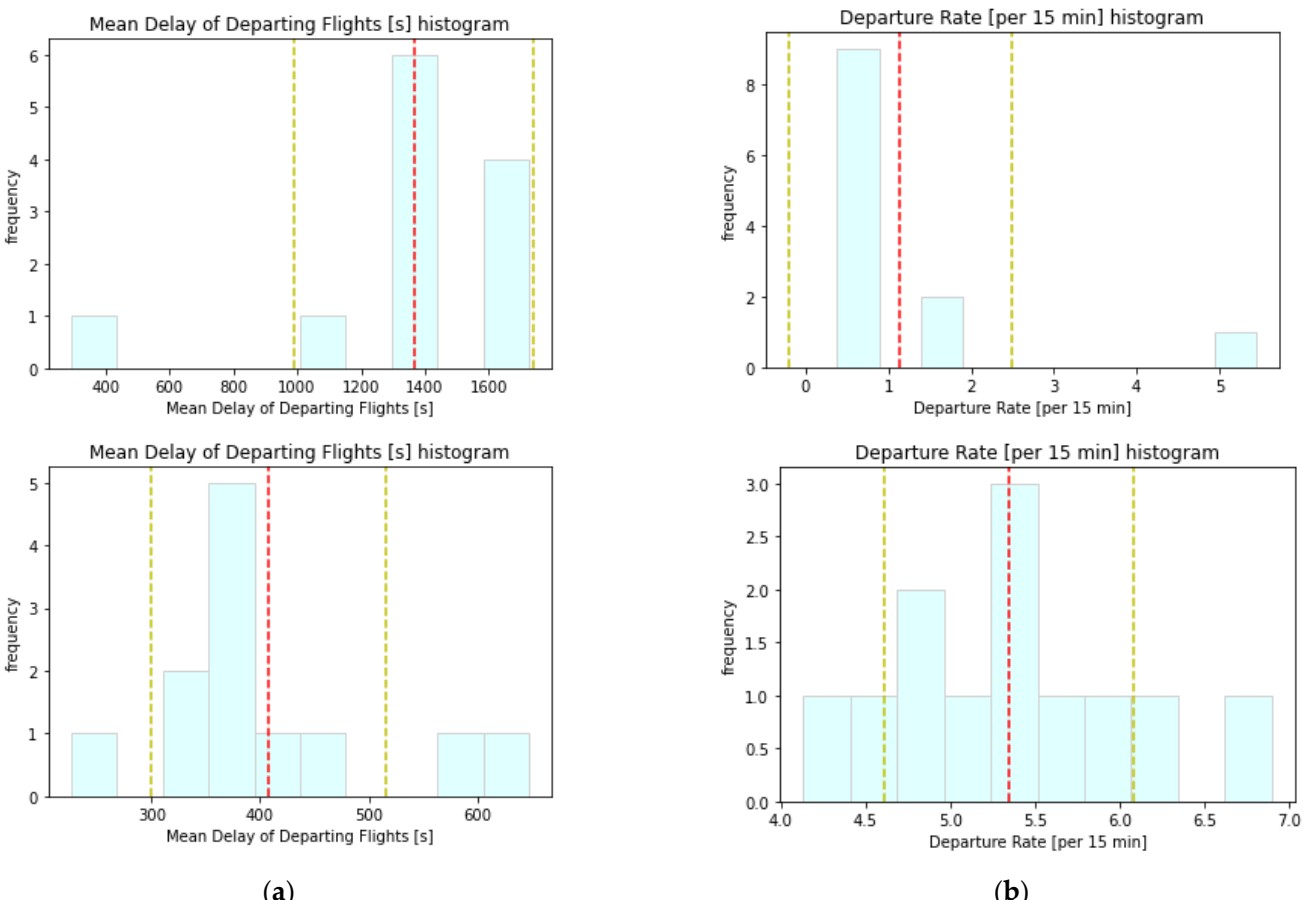

(**a**)                                                                                                                        (**b**)

**Figure 19.** Simulation results. (**a**) Histograms of mean delay of departing flights for reference scenarios (**up**) and solution scenarios (**down**). (**b**) Histograms of departure rate (per 15 min) for reference scenarios (**up**) and solution scenarios (**down**). Red vertical lines represent the mean values, whereas yellow vertical lines represent the intervals for the standard deviation.

*5.2. Risk Assessment Results*

Our previous work [14] developed several detailed event trees for threat scenarios regarding the intrusion of a drone in the departure path of the runways at LIMC. Figure 20 shows two event trees for departures on runways 35R or 35L: the former for an intrusion taking place less than 1 km from the airport; the latter for an intrusion taking place at a greater distance.

The event trees highlight in pink the branches that represent an unacceptable risk. In particular, for the first event tree, the following two branches are unacceptable for the intrusion that occurred close to the airport: Branch 2 and Branch 6, which both involve the loss of efficiency of the mitigation (i.e., neutralization) capability. Instead, for the second event tree, the following three branches pose unacceptable risks: Branch 5, Branch 6, and Branch 8. They are all related to a loss of efficiency in the detection means. In addition to this condition, Branch 6 involves a loss of efficiency in the mitigation capability, whereas

Branch 8 involves a loss of efficiency in all the technical means. See [14] for further details about the estimation of the probabilities and the identification of the unacceptable branches.

| Bid | Pts | Pd | Pa | Pm | BProba |
|---|---|---|---|---|---|
| 1 | | | | 0.6 | $1.8\times10^{-3}$ |
| 2 | | | 0.99 | 0.4 | $1.2\times10^{-3}$ |
| 3 | | | 0.01 | 0.6 | $1.8\times10^{-5}$ |
| 4 | $3.9\times10^{-3}$ | 0.8 | | 0.4 | $1.2\times10^{-5}$ |
| 5 | | 0.2 | | 0.6 | $4.6\times10^{-4}$ |
| 6 | | | 0.99 | 0.4 | $3.1\times10^{-4}$ |
| 7 | | | 0.01 | 0.6 | $4.6\times10^{-6}$ |
| 8 | | | | 0.4 | $3.1\times10^{-6}$ |

(**a**)

| Bid | Pts | Pd | Pa | Pm | BProba |
|---|---|---|---|---|---|
| 1 | | | | 0.6 | $9.9\times10^{-4}$ |
| 2 | | | 0.99 | 0.4 | $6.6\times10^{-4}$ |
| 3 | | | 0.01 | 0.6 | $1.0\times10^{-5}$ |
| 4 | $3.9\times10^{-3}$ | 0.43 | | 0.4 | $6.7\times10^{-6}$ |
| 5 | | 0.57 | | 0.6 | $1.3\times10^{-3}$ |
| 6 | | | 0.99 | 0.4 | $8.7\times10^{-4}$ |
| 7 | | | 0.01 | 0.6 | $1.3\times10^{-5}$ |
| 8 | | | | 0.4 | $8.8\times10^{-6}$ |

(**b**)

**Figure 20.** Event trees for the intrusion of an unauthorized drone in LIMC 35 departure path when takeoff is occurring [14]. (**a**) <1 km from runway. (**b**) >1 km from runway.

Compared to these event trees, some preliminary considerations may be introduced for the ASPRID operational concept with regard to safety aspects. In detail, the following results have been achieved for the simulated solution scenarios:

- ASPRID's operational concept has always efficiently detected the intruder drone in the given threat scenarios. This contributes to a potential de-risking of all the unacceptable branches in Figure 20b.
- With regard to threat assessments, no consideration may be provided since the successful/unsuccessful assessments of intruder drones have not been modeled in the real-time simulator.
- ASPRID's operational concept has always neutralized the intruder drone in the given threat scenarios, where allowed by the countermeasures' coverage. In case a neutralization was not available due to the intruder's trajectory (i.e., in the scenarios "threat parallel to runways" and "threat on routes—Drone 01"), the ASPRID operational procedure ruled out safety issues. This is confirmed by the results in Table A2, which show that, with regard to the abovementioned scenarios: (i) no collisions and no red alerts occurred; (ii) the safety degradation was only major (2). This contributes to a potential de-risking of Branch 2 and Branch 6 in Figure 20a.

Clearly, more extensive simulations are required to confirm these preliminary considerations about the safety impacts of the ASPRID operational concept and its de-risking effect for unacceptable branches in ETA. Moreover, such considerations are valid in the case of the operational environment used for the simulation of solution scenarios, which may be interpreted as a set of requirements in order to reach the estimated values for the performance metrics. In other words, the proposed approach enables reverse reasoning to allocate the required reliability to the technical means for detection, assessment, and neutralization.

## 6. Discussion

This section provides a discussion of the achieved results for the proposed case study. Such results demonstrate the resilience-driven effectiveness of the ASPRID operational concept to protect airport operations against drone intrusions. Indeed, the proposed evaluation methodology has assessed the main performance benefits related to the increase in the airport's resilience to drone intrusions by means of the ASPRID concept. These benefits may be summarized in the following key points in the context of drone intrusions:

- Guarantee the safe continuity of airport operations in normal or degraded modes—The safety of airport operations is preserved, as demonstrated by the simulation results

for the safety metrics. Moreover, the ASPRID system potentially contributes to the de-risking of most critical branches in the event trees, as preliminarily discussed in regard to risk assessment results. These numbers are reached despite that the closure of the overall airport was avoided in all the solution scenarios by means of the ASPRID operational procedure, which limited the interruption to those operations that are strictly affected by the intruder.

- Reduce time performance and capacity degradations—This benefit is demonstrated by the simulation results for time performance and capacity metrics, which report a clear improvement with respect to reference scenarios. Clearly, such a benefit directly implies an increase in airport resilience in the face of drone intrusions.
- Limit the increase in workload—An increase in the human workload is observed for solution scenarios, as expected. Anyway, even if this does not really represent a performance benefit, the workload increase is slight with respect to reference scenarios. Additional mitigation of this increase may be further explored in future works.
- Improve the situational awareness and early-management capabilities of airport operations with respect to intrusions—This benefit is not directly measured, but it is tangible due to the alerting functions. Moreover, the previous benefits are a clear consequence of this one.
- Lastly, it is worth noting that, with respect to reference scenarios, the assessed improvement in solution scenarios is even greater in the case of long-lasting drone intrusions.

## 7. Conclusions

This paper deals with the protection of airport operations against drone intrusions. It proposes a resilience-driven operational concept for such protection in order to minimize the performance degradation of airport operations by avoiding the closure of the overall airport if not strictly necessary. The operational concept is modeled in terms of a process model and an operational procedure. Moreover, a resilience-driven evaluation is proposed to assess the performance benefits of the concept. The evaluation exploits a combination of real-time simulations and risk analysis activities on event trees. The proposed evaluation is applied to a concrete case study related to the Milan Malpensa Airport.

Several research directions will be considered for future work. Firstly, the evaluation may be extended by also exploiting fast-time simulation tools, in order to perform the extensive simulations that are required to study the impact of all the control variables on the concept's effectiveness for managing threat scenarios. Then, real-time simulations may be enriched with simulation campaigns involving several controllers and experts (e.g., representing LEAs, aerodrome safety and security operators, etc.), in order to learn more about the sensitivity of real-time simulation results with respect to real users. In detail, a simulation-based gaming exercise has already been performed [36] and will be used to complete the assessment of the reference metrics by considering the users' feedback. Finally, the operational concept may be further refined to consider different improvements, including: variable-size safety bubbles for alerting, according to the features of intruder drones (e.g., speed); mitigation of the increase in workload to perform the operational procedure in case of intrusions; and management of additional types of drone intrusions, such as swarm intrusions or cyber drone intrusions.

**Author Contributions:** Conceptualization, D.P., G.G., A.V., M.S., M.I., P.B., T.D. and E.M.; data curation, D.P., M.S. and M.I.; formal analysis, D.P., G.G., P.B. and T.D.; funding acquisition, M.I., A.V., P.B. and E.M.; methodology, D.P., G.G., P.B. and T.D.; software, D.P., G.G. and M.S.; validation, D.P. and M.S.; visualization, D.P., M.S., P.B. and T.D.; writing—original draft, D.P., G.G., M.S., P.B. and T.D.; writing—review and editing, D.P., G.G., A.V., M.S., M.I., P.B., T.D. and E.M. All authors have read and agreed to the published version of the manuscript.

**Funding:** This research was funded by the SESAR Joint Undertaking, grant agreement No 892036, ASPRID project (Airport System PRotection from Intruding Drones).

**Data Availability Statement:** Publicly available datasets were generated and analyzed in this study related to the configuration and the logs of the real-time simulation exercises. This data can be found here: https://github.com/MaurizioSodano/asprid-file-storage (accessed on 7 April 2023).

**Acknowledgments:** This research has been started within GARTEUR Aviation Security Group of Responsables.

**Conflicts of Interest:** The authors declare no conflict of interest. The funders had no role in the design of the study; in the collection, analyses, or interpretation of data; in the writing of the manuscript; or in the decision to publish the results.

## Appendix A

This appendix reports the detailed results about the simulation activities for the evaluation of the ASPRID operational concept. The results regard the estimation of the performance metrics for both reference scenarios and solution scenarios. The metrics are estimated for each threat scenario. The following tables (Tables A1–A5) list the detailed results for each metric set (safety, time performance, workload, and capacity).

**Table A1.** Simulation results: safety metrics of reference scenarios. The following colors are used for SDM values: red for catastrophic (4) value; orange for hazardous (3) value; yellow for major (2) value; green for minor (1) value. Bold values respectively represent the mean and the standard deviation of SDM values.

| Threat Scenario | NYI | NOI | NRI | MTYI (s) | MTOI (s) | MTRI (s) | NYI-P | NOI-P | NRI-P | NYI-A | NOI-A | NRI-A | SDM |
|---|---|---|---|---|---|---|---|---|---|---|---|---|---|
| Threat on 35L—Drone 01 | 8 | 3 | 2 | 74.38 | 71.00 | 1906.50 | 2 | 6 | 0 | 3 | 0 | 2 | **4** |
| Threat on 35L—Drone 02 | 12 | 5 | 2 | 772.80 | 445.78 | 1011.45 | 3 | 9 | 0 | 5 | 0 | 2 | **4** |
| Threat on 35L—Drone 03 | 5 | 4 | 1 | 51.40 | 1544.00 | 2047.00 | 0 | 5 | 0 | 4 | 0 | 1 | **4** |
| Threat on 35R—Drone 01 | 5 | 4 | 1 | 418.58 | 1332.18 | 1760.90 | 1 | 4 | 0 | 4 | 0 | 1 | **4** |
| Threat on 35R—Drone 02 | 2 | 2 | 1 | 36.00 | 1102.00 | 2188.00 | 0 | 2 | 0 | 2 | 0 | 1 | **4** |
| Threat on 35R—Drone 03 | 4 | 2 | 1 | 1076.85 | 1098.35 | 2103.70 | 0 | 4 | 0 | 2 | 0 | 1 | **4** |
| Threat on fingers—Drone 01 | 10 | 6 | 2 | 1178.08 | 970.48 | 41.50 | 3 | 7 | 0 | 6 | 0 | 2 | **4** |
| Threat parallel to runways—Drone 01 | 8 | 0 | 0 | 591.35 | 0.00 | 0.00 | 1 | 7 | 0 | 0 | 0 | 0 | **4** |
| Threat parallel to runways—Drone 04 | 3 | 0 | 0 | 198.67 | 0.00 | 0.00 | 0 | 3 | 0 | 0 | 0 | 0 | **4** |
| Threat on routes—Drone 01 | 2 | 0 | 0 | 2033.50 | 0.00 | 0.00 | 0 | 2 | 0 | 0 | 0 | 0 | **4** |
| Threat on routes—Drone 03 | 2 | 2 | 0 | 159.50 | 1976.50 | 0.00 | 0 | 2 | 0 | 2 | 0 | 0 | **4** |
| Threat on tower—Drone 01 | 11 | 4 | 1 | 1374.64 | 22.75 | 12.00 | 3 | 8 | 0 | 4 | 0 | 1 | **4** |
| Mean | 6.00 | 2.67 | 0.92 | 663.81 | 713.59 | 922.59 | 1.08 | 4.92 | 0.00 | 2.67 | 0.00 | 0.92 | **4.00** |
| Standard Deviation | 3.51 | 1.93 | 0.76 | 611.08 | 678.33 | 955.23 | 1.26 | 2.36 | 0.00 | 1.93 | 0.00 | 0.76 | **0.00** |

**Table A2.** Simulation results: safety metrics of solution scenarios. The following colors are used for SDM values: red for catastrophic (4) value; orange for hazardous (3) value; yellow for major (2) value; green for minor (1) value. Bolds values respectively represent the mean and the standard deviation of SDM values.

| Threat Scenario | NYI | NOI | NRI | MTYI (s) | MTOI (s) | MTRI (s) | NYI-P | NOI-P | NRI-P | NYI-A | NOI-A | NRI-A | SDM |
|---|---|---|---|---|---|---|---|---|---|---|---|---|---|
| Threat on 35L—Drone 01 | 3 | 2 | 0 | 183.00 | 333.00 | 0.00 | 0 | 3 | 0 | 2 | 0 | 0 | **2** |
| Threat on 35L—Drone 02 | 10 | 2 | 0 | 284.50 | 263.00 | 0.00 | 4 | 6 | 0 | 2 | 0 | 0 | **2** |
| Threat on 35L—Drone 03 | 3 | 2 | 0 | 347.00 | 700.00 | 0.00 | 0 | 3 | 0 | 2 | 0 | 0 | **2** |

**Table A2.** *Cont.*

| Threat Scenario | NYI | NOI | NRI | MTYI (s) | MTOI (s) | MTRI (s) | NYI-P | NOI-P | NRI-P | NYI-A | NOI-A | NRI-A | SDM |
|---|---|---|---|---|---|---|---|---|---|---|---|---|---|
| Threat on 35R—Drone 01 | 2 | 1 | 0 | 221.00 | 293.00 | 0.00 | 0 | 2 | 0 | 1 | 0 | 0 | 2 |
| Threat on 35R—Drone 02 | 1 | 1 | 0 | 43.00 | 410.00 | 0.00 | 0 | 1 | 0 | 1 | 0 | 0 | 2 |
| Threat on 35R—Drone 03 | 3 | 2 | 0 | 139.00 | 615.50 | 0.00 | 1 | 2 | 0 | 2 | 0 | 0 | 2 |
| Threat on fingers—Drone 01 | 15 | 7 | 2 | 93.87 | 47.57 | 41.50 | 5 | 10 | 1 | 6 | 0 | 2 | 3 |
| Threat parallel to runways—Drone 01 | 11 | 0 | 1 | 161.19 | 0.00 | 188.00 | 1 | 10 | 0 | 0 | 0 | 0 | 2 |
| Threat parallel to runways—Drone 04 | 16 | 0 | 2 | 236.23 | 0.00 | 249.00 | 4 | 12 | 0 | 0 | 0 | 0 | 2 |
| Threat on routes—Drone 01 | 2 | 0 | 0 | 430.45 | 0.00 | 0.00 | 0 | 2 | 0 | 0 | 0 | 0 | 1 |
| Threat on routes—Drone 03 | 2 | 2 | 0 | 374.50 | 695.50 | 0.00 | 0 | 2 | 0 | 2 | 0 | 0 | 2 |
| Threat on tower—Drone 01 | 7 | 2 | 0 | 427.29 | 481.50 | 0.00 | 2 | 5 | 0 | 2 | 0 | 0 | 2 |
| Mean | 6.25 | 1.75 | 0.42 | 245.09 | 319.92 | 39.88 | 1.42 | 4.83 | 0.08 | 1.67 | 0.00 | 0.17 | **2.00** |
| Standard Deviation | 5.18 | 1.79 | 0.76 | 123.41 | 256.98 | 81.64 | 1.80 | 3.65 | 0.28 | 1.55 | 0.00 | 0.55 | **0.41** |

**Table A3.** Simulation results: time performance metrics. The following colors are used for TDM values: red for high (2) value; orange for medium (1) value; green for low (0) value. Bold values respectively represent the mean and the standard deviation of TDM values.

| Threat Scenario | Reference Scenarios | | | | | | | Solution Scenarios | | | | | | |
|---|---|---|---|---|---|---|---|---|---|---|---|---|---|---|
| | Departures | Takeoffs | NDDF | PDDF (%) | MDDF (s) | WDDF (s) | TDM | Departures | Takeoffs | NDDF | PDDF (%) | MDDF (s) | WDDF (s) | TDM |
| Threat on 35L—Drone 01 | 11 | 2 | 10 | 90.91 | 1328.81 | 2071.12 | 2 | 15 | 7 | 8 | 53.33 | 400.29 | 568.57 | 0 |
| Threat on 35L—Drone 02 | 9 | 2 | 8 | 88.89 | 1358.29 | 2071.12 | 2 | 14 | 6 | 8 | 57.14 | 393.38 | 739.22 | 0 |
| Threat on 35L—Drone 03 | 8 | 2 | 7 | 87.50 | 1423.03 | 2071.12 | 2 | 17 | 10 | 8 | 47.06 | 382.44 | 650.01 | 0 |
| Threat on 35R—Drone 01 | 11 | 4 | 9 | 81.82 | 1085.75 | 2071.12 | 1 | 13 | 6 | 6 | 46.15 | 225.54 | 472.10 | 0 |
| Threat on 35R—Drone 02 | 8 | 1 | 7 | 87.50 | 1720.18 | 2596.10 | 2 | 11 | 4 | 4 | 36.36 | 355.15 | 612.10 | 0 |
| Threat on 35R—Drone 03 | 8 | 1 | 7 | 87.50 | 1725.79 | 2596.10 | 2 | 17 | 11 | 11 | 64.71 | 384.96 | 736.40 | 0 |
| Threat on fingers—Drone 01 | 9 | 2 | 8 | 88.89 | 1381.82 | 2071.12 | 2 | 15 | 9 | 7 | 46.67 | 336.19 | 538.22 | 0 |
| Threat parallel to runways—Drone 01 | 9 | 2 | 8 | 88.89 | 1371.04 | 2071.12 | 2 | 13 | 5 | 6 | 46.15 | 459.20 | 1111.07 | 0 |
| Threat parallel to runways—Drone 04 | 15 | 8 | 7 | 46.67 | 292.14 | 546.10 | 0 | 19 | 11 | 14 | 73.68 | 391.65 | 948.81 | 0 |
| Threat on routes—Drone 01 | 9 | 2 | 8 | 88.89 | 1357.77 | 2071.12 | 2 | 19 | 12 | 12 | 63.16 | 648.00 | 1184.22 | 1 |
| Threat on routes—Drone 03 | 9 | 1 | 8 | 88.89 | 1626.39 | 2596.10 | 2 | 17 | 10 | 12 | 70.59 | 582.62 | 1161.42 | 0 |
| Threat on tower—Drone 01 | 8 | 1 | 7 | 87.50 | 1718.46 | 2596.10 | 2 | 15 | 8 | 7 | 46.67 | 331.52 | 602.12 | 0 |
| Mean | 9.50 | 2.33 | 7.83 | 84.49 | 1365.79 | 2119.03 | **1.75** | 15.42 | 8.25 | 8.58 | 54.31 | 407.58 | 777.02 | **0.08** |
| Standard Deviation | 1.94 | 1.89 | 0.90 | 11.59 | 374.57 | 532.33 | **0.60** | 2.36 | 2.52 | 2.87 | 11.03 | 107.83 | 245.82 | **0.28** |

**Table A4.** Simulation results: workload metrics. The following colors are used for WDM values: red for high (2) value; orange for medium (1) value; green for low (0) value. Bold values respectively represent the mean and the standard deviation of WDM values.

| Threat Scenario | Reference Scenarios | | | | Solution Scenarios | | | |
|---|---|---|---|---|---|---|---|---|
| | Departures | NCDF | NNCDF | WDM | Departures | NCDF | NNCDF | WDM |
| Threat on 35L—Drone 01 | 11 | 92 | 8.36 | 1 | 15 | 71 | 4.73 | 0 |
| Threat on 35L—Drone 02 | 9 | 39 | 4.33 | 0 | 14 | 64 | 4.57 | 0 |
| Threat on 35L—Drone 03 | 8 | 36 | 4.50 | 0 | 17 | 105 | 6.18 | 1 |
| Threat on 35R—Drone 01 | 11 | 49 | 4.45 | 0 | 13 | 56 | 4.31 | 0 |
| Threat on 35R—Drone 02 | 8 | 34 | 4.25 | 0 | 11 | 44 | 4.00 | 0 |
| Threat on 35R—Drone 03 | 8 | 34 | 4.25 | 0 | 17 | 91 | 5.35 | 1 |
| Threat on fingers—Drone 01 | 9 | 44 | 4.89 | 0 | 15 | 85 | 5.67 | 1 |

**Table A4.** *Cont.*

| Threat Scenario | Reference Scenarios | | | | Solution Scenarios | | | |
|---|---|---|---|---|---|---|---|---|
| | Departures | NCDF | NNCDF | WDM | Departures | NCDF | NNCDF | WDM |
| Threat parallel to runways—Drone 01 | 9 | 38 | 4.22 | 0 | 13 | 70 | 5.38 | 1 |
| Threat parallel to runways—Drone 04 | 15 | 99 | 6.60 | 1 | 19 | 109 | 5.74 | 1 |
| Threat on routes—Drone 01 | 9 | 38 | 4.22 | 0 | 19 | 104 | 5.47 | 1 |
| Threat on routes—Drone 03 | 9 | 37 | 4.11 | 0 | 17 | 88 | 5.18 | 1 |
| Threat on tower—Drone 01 | 8 | 33 | 4.13 | 0 | 15 | 77 | 5.13 | 1 |
| Mean | 9.50 | 47.75 | 4.86 | **0.17** | 15.42 | 80.33 | 5.14 | **0.67** |
| Standard Deviation | 1.94 | 21.83 | 1.24 | **0.37** | 2.36 | 19.51 | 0.61 | **0.47** |

**Table A5.** Simulation results: capacity metrics.

| Threat Scenario | Reference Scenarios | | Solution Scenarios | |
|---|---|---|---|---|
| | Departures | DR | Departures | DR |
| Threat on 35L—Drone 01 | 11 | 1.50 | 15 | 4.44 |
| Threat on 35L—Drone 02 | 9 | 0.76 | 14 | 4.92 |
| Threat on 35L—Drone 03 | 8 | 0.75 | 17 | 5.31 |
| Threat on 35R—Drone 01 | 11 | 1.50 | 13 | 6.90 |
| Threat on 35R—Drone 02 | 8 | 0.38 | 11 | 6.14 |
| Threat on 35R—Drone 03 | 8 | 0.38 | 17 | 5.49 |
| Threat on fingers—Drone 01 | 9 | 0.75 | 15 | 6.06 |
| Threat parallel to runways—Drone 01 | 9 | 0.75 | 13 | 4.96 |
| Threat parallel to runways—Drone 04 | 15 | 5.45 | 19 | 4.13 |
| Threat on routes—Drone 01 | 9 | 0.75 | 19 | 5.26 |
| Threat on routes—Drone 03 | 9 | 0.38 | 17 | 4.92 |
| Threat on tower—Drone 01 | 8 | 0.38 | 15 | 5.63 |
| Mean | 9.50 | 1.14 | 15.42 | 5.35 |
| Standard Deviation | 1.94 | 1.35 | 2.36 | 0.73 |

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
