# Peer review of "Evaluation of a Resilience-Driven Operational Concept to Manage Drone Intrusions in Airports"

_information, doi:10.3390/info14040239_

Round 1
Reviewer 1 Report
This is a piece of good work. I, however, have two comments.
1. it would have been good if there was a section 2.3. with a summary of the literature review and an explanation of the motivation for the research and an identification of the direct research gap that the presented research paper solves. Currently, this is not specified all that well.
2. the BPMN diagrams are not structured correctly. Every process should have a start and stop object on the diagram. Drawings that include BPMN should be improved.
Reviewer 2 Report
This paper evaluates resilience-driven operational concept about the protection of airport operations against drone intrusions. The evaluation is based on real-time simulations as well as event tree analysis. In authors' previous work, they assessed risk of airport drone intrusions with a unified methodological framework, now they are evaluating those concepts with real-time simulation in a case study related to Milan Malpensa airport. The simulation results demonstrate the benefits related to the increase of airport resilience to drone intrusions by adapting ASPRID concept.
The topic is interesting and important, I am curious if the concepts already be taken into consideration by the airports?
